# Self-Explained Keywords Empower Large Language Models for Code Generation

## Abstract

Large language models (LLMs) have achieved impressive performance in code generation. Despite the remarkable success, we observed that LLMs often misunderstand or overlook some problem-specific undertrained keywords during code generation, compromising the accuracy of the generated code. After explicitly explaining these undertrained keywords using well-trained terms in the prompt, LLMs are more likely to generate correct code implementation. Inspired by this observation, we propose a novel technique named SEK (**S**elf-**E**xplained **K**eywords), which empowers an LLM for better code generation by extracting and explaining the key terms in the problem description with the LLM itself. Comprehensive experiments across three benchmarks, i.e., HumanEval(+), MBPP(+), and APPS, with five representative LLMs, show that SEK can significantly improve LLMs in code generation, yielding substantial and consistent gains. For instance, SEK improves the Pass@1 of DeepSeek-Coder-V2-Instruct from 85.4% to 93.3% on the Humaneval benchmark. Further analysis confirms that SEK enables the LLMs to shift their attention from low-frequency keywords to their corresponding high-frequency counterparts.

## 1 Introduction

Code generation aims to generate a code snippet that meets the intent described in natural language. This process can potentially reduce the costs of software development (Xu et al., 2022; Yin & Neubig, 2017; Vaithilingam et al., 2022). Recently, the notable success of LLMs such as ChatGPT (OpenAI, 2022) and Llama-3 (AI@Meta, 2024) has substantially enhanced the state-of-the-art in code generation. These LLMs demonstrate remarkable proficiency in comprehending natural language descriptions and translating them into code snippets.

Despite the remarkable success, we found that LLMs often struggle to translate certain terms in the problem description into corresponding code. When these terms are critical in the programming context (i.e., serving as a *keyword*), this limitation can compromise the accuracy of the generated code. An example is presented in Figure 1, where the coding problem requires returning *even digits* within a given range in ascending order. We found that LLMs fail to recognize that this term refers to the even numbers *between 0 and 9*, leading to the omission of this constraint in the generated conditional statements. One possible reason for this observation is the long-tail distribution of coding training datasets (Chen et al., 2024d; Zhong et al., 2024b), where some programming terms are rare and undertrained and thus cannot be effectively translated into the corresponding code by the LLM. If we explicitly convert *even digits* into well-trained terms by explaining it and prompt the LLM to focus on it, the LLM can produce a correct implementation.

Inspired by this example, we hypothesize that we can boost LLMs for code generation by explicitly identifying and explaining certain keywords. However, this is non-trivial and usually requires manual efforts. Our key idea is that such keywords can be identified and explained by LLM *themselves*. This idea is supported by three observations: (1) prior studies show that LLMs can effectively identify task-specific key items (Fang et al., 2024; Fan et al., 2024); (2) our experiments indicate that such LLM-selected keywords are often terms that are more likely to be undertrained, i.e., have a relatively low frequency in the code training set (detailed in Appendix E.5); and (3) although the direct mapping from these keywords to code may be undertrained, the semantics of these keywords

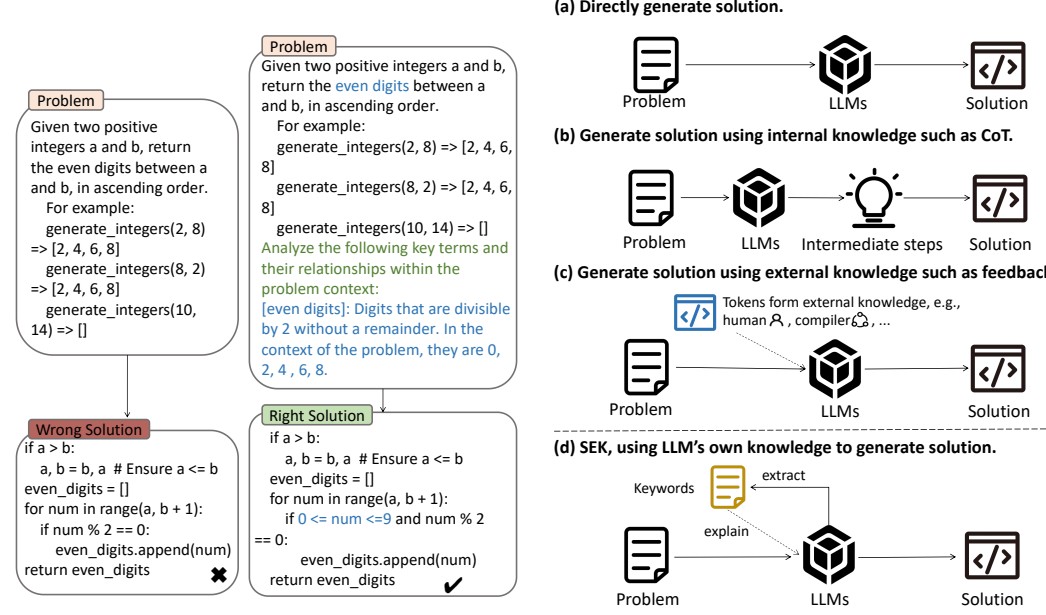

Figure 1: Motivating example.

Figure 2: Schematic illustration of various generation approaches.

are typically understandable by LLMs after pre-training on large-scale general corpora. This enables LLMs to describe and explain these keywords using natural language.

Based on this idea, this work proposes **S**elf-**E**xplained **K**eyword (SEK), a novel technique leveraging LLMs' own comprehension capabilities to automatically identify and explain these problem-specific keywords to enhance their understanding of coding problems. SEK employs a carefully designed prompt with a few examples, directing LLMs to focus on crucial keywords in the problem description. We use a frequency-based ranking algorithm to sort these keywords and further prioritize low-frequency keywords, which are then appended to the original problem description to construct an augmented prompt. Overall, this approach aligns with the working process of pragmatic developers, which use auxiliary tools like blackboards to highlight, explain, and rank important parts of requirements (Andrew Hunt, 2000).

SEK enhances LLMs' problem-solving capabilities in a novel way, distinguishing itself from previous methods in prompt engineering for code generation. As shown in Figure 2, unlike previous approaches that often rely on introducing external knowledge, such as human feedback (Chen et al., 2023a; Wu et al., 2024; Dubois et al., 2024) or the execution results of LLM-generated solutions (Zhong et al., 2024b; Chen et al., 2023c; Zhong et al., 2024a), into the input, SEK operates by distilling additional content from the problem description using the LLM itself. Chain of Thought (CoT) (Wei et al., 2022), which also utilizes LLMs' inherent knowledge for problem-solving, bears the closest resemblance to SEK. However, the fundamental strategies of CoT and SEK are different: CoT guides the LLM to think in a chain-like manner, while SEK directs the LLM to understand and prioritize key concepts.

We evaluate SEK with five representative LLMs, including three open-source models and two closed-source models, on three widely used code generation benchmarks. Experimental results demonstrate that SEK effectively enhances code generation performance. For example, SEK enables Llama-3.1 to achieve a relative improvement of 8.8% averaged on the used benchmarks. Notably, DeepSeek-Coder-V2-Instruct with SEK significantly outperforms it with standard prompting, achieving state-of-the-art performance on several benchmarks (e.g., HumanEval: 85.4% to 93.3%). Furthermore, our ablation studies indicate that the carefully designed prompt and the ranking component of SEK are effective. Additionally, our attention analysis reveals that SEK helps LLMs comprehend low-frequency keywords by redirecting attention to their high-frequency counterparts. Comparative case studies with other baselines further illustrate SEK's efficacy in enhancing LLMs'

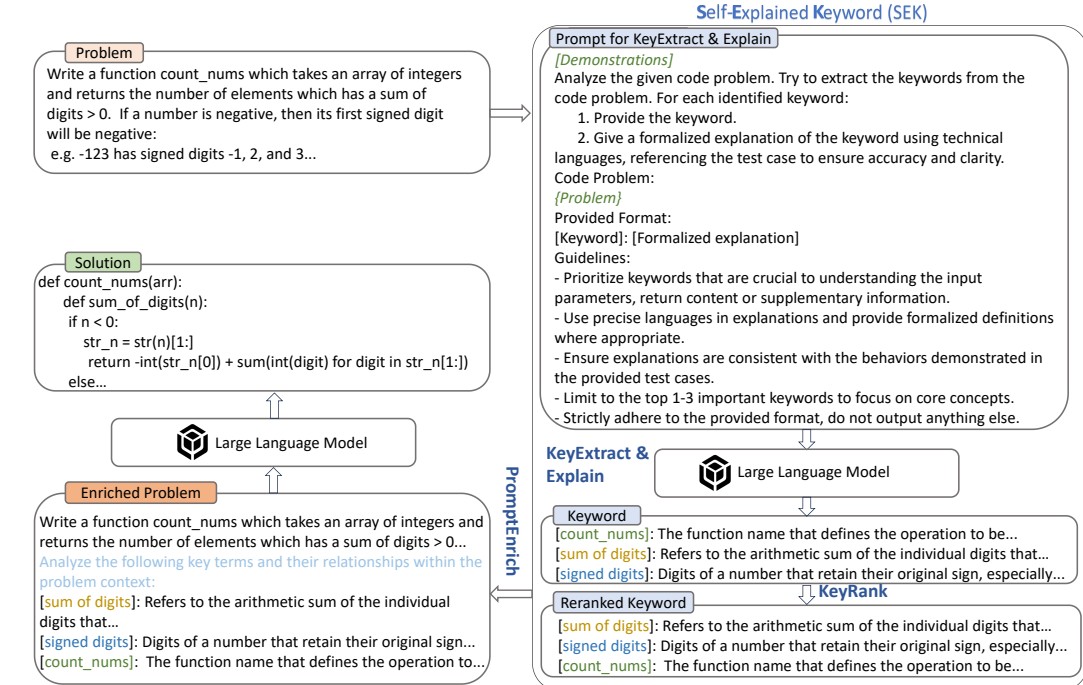

Figure 3: The overview of Self-Explained Keyword. The details in each step are omitted.

understanding of low-frequency, problem-specific keywords. Our code is in the Supplementary Materials and will be made public after review.

## 2 METHODOLOGY

Code generation aims to generate a solution program based on a problem description. Typically, a problem description includes implementation requirements, and several test cases to help further understand the problem.

Figure 3 illustrates the overview of SEK. SEK is designed to address the issue of LLMs overlooking low-frequency terms in the program description due to the long-tail distribution in their training data. To address it, one key is to leverage the LLM's capabilities to identify and explain potentially overlooked keywords within the problem description. We employ a carefully crafted prompt with a few-shot learning method to achieve this. After obtaining the keywords and their explanations, another challenge is how to effectively integrate them with the original problem description. For this purpose, we introduce a frequency-based ranking algorithm that prioritizes less frequent tokens, which are more likely to be overlooked by the LLM. These ranked keywords are then appended to the original problem description, serving to guide the LLM towards generating an accurate solution. The process comprises three main steps:

**KeyExtract & Explain** (Section 2.1): Based on the problem description, SEK constructs a prompt to guide the LLM to identify and explain keywords within the problem description.

**KeyRank** (Section 2.2): SEK employs a frequency-based ranking algorithm to prioritize the extracted keywords.

**PromptEnrich** (Section 2.3), SEK concatenates the ranked keywords and their explanations with the original problem description to create an enriched problem description. This comprehensive formulation serves as the final input for the LLM to generate code solutions.

## 2.1 KEYEXTRACT & EXPLAIN

In this step, SEK extracts and explains keywords from the given problem description. Our key insight is that LLMs inherently possess strong understanding and reasoning abilities after training on large-scale general corpora, enabling them to explain crucial concepts within a problem description. The effectiveness of using LLMs for keyword extraction has also been demonstrated by recent studies (Maragheh et al., 2023; Lee et al., 2023). Inspired by this insight, SEK uses the LLM itself to perform the task with a prompt-based approach.

Specifically, SEK begins by designing a prompt to instruct an LLM for keyword extraction and explanation. The prompt is shown in *Prompt for KeyExtract & Explain* in Figure 3, which consists of three parts. First, it provides the overall instruction for the task, namely the generation of keywords and their corresponding explanations. Then, it specifies the format of input and output. Finally, it provides detailed guidelines. Intuitively, terms associated with input, output, and supplementary content (i.e., clarifications of keywords or specifications of value ranges) within the problem description are relatively important, as they contain the problem's core elements, objectives, and constraints (Guideline 1). For explanations, given the potential ambiguity in natural language expressions and the clarity of the public test cases, the generated explanations should be both precise and consistent with these test cases (Guideline 2,3). We also impose limitations on the keyword quantity to guarantee that the LLM identifies and outputs only the important keywords in the problem description (Guideline 4). The LLM is prompted to identify at most three keywords and generate an explanation for each identified keyword. Ultimately, to facilitate subsequent processing, we further emphasize the output format (Guideline 5). Additionally, we use several examples to leverage LLMs' in-context learning ability to understand and solve this task.

## 2.2 KEYRANK

After extracting and explaining the keywords, the next goal is to enhance the original prompt. Previous research has demonstrated that LLMs are sensitive to the order of tokens in the prompt, known as position bias (Li et al., 2024; Yu et al., 2024). It highlights the need to carefully arrange the extracted keywords. Notably, pragmatic human developers tend to place more important keywords at the beginning in practice (Andrew Hunt, 2000). This preference may be reflected in the training dataset, leading LLMs to also focus more on the keywords written at the front. Therefore, we propose a set of heuristic rules to rank keywords by importance, namely **KeyRank**. The specific Algorithm is provided in the Appendix A.

We first examine the keywords extracted by two LLMs (Llama 3.1 and DeepSeekCoder-V2) for part of the coding problems in the APPS training set. These keywords can generally be categorized into three types: (1) *Function keywords*, which match the desired function names, such as `count_nums` in Figure 3. (2) *General keywords*, which appear in the problem description, like `sum of digits` in Figure 3. (3) *Abstract keywords*, which do not appear in any input; instead, they are abstract terms summarized from multiple concepts. For example, for two different concepts "substring before the dot" and "substring after the dot" in the problem description, LLM may combine them into a single keyword `substring before/after the dot`. The proportions of these three categories are 22.5%, 59.9%, and 17.7%.

We hypothesize that abstract keywords are the most important, as they encompass explanations across multiple concepts. General keywords refer to single concepts and are of secondary importance, while function Keywords, whose explanations have already appeared in the problem description, are the least important. Therefore, we propose ordering the keywords as *abstract → general → function*. Appendix E.1 demonstrates that this heuristic combination order yields the best results.

Moreover, since general keywords represent the majority (59.9%) and LLMs could extract multiple general keywords for a single problem, we further perform an internal ranking of these general keywords. We argue that a keyword is more important if it appears more frequently in the problem description (i.e., higher term frequency). Conversely, if a keyword appears less frequently in a corpus (i.e., lower document frequency), the corresponding code conversion could be more challenging as we stated in the Introduction section, and thus its explanation is more significant. Therefore, we use the TF-IDF, a widely used metric that combines term frequency (TF) and inverse document frequency (IDF), to assess the importance of general keywords. TF-IDF is calculated as follows:

$$\text{TF-IDF} = \frac{n_i}{\sum_k n_k} \times \log \frac{|D|}{1 + |\{j : t_i \in d_j\}|}.$$

The first term represents TF, where $n_i$ denotes the number of times the keyword appears in the problem description, and the denominator represents the total occurrences of all items with the same number of grams. The second term represents IDF, where $|D|$ is the total number of documents in the corpus, and the denominator represents the number of documents containing the keyword $t_i$.

We adopt the Python subset of the eval-codealpaca-v1 (Luo et al., 2023) as the corpus for computing document frequency, which is generated by ChatGPT and can partially reflect the distribution of LLMs' training data. In addition, we demonstrate that SEK is robust across various corpora.

### 2.3 PROMPTENRICH

After obtaining the ranked keywords and their explanations, SEK integrates them with the original problem. As shown in the enriched problem in Figure 3, SEK appends the ranking results to the end of the problem, providing additional explanations for key concepts in the problem. It's worth noting that, to maintain the coherence of the problem context, we insert the phrase *"Analyze the following key terms and their relationships within the problem context:"* after the problem. This acts as a semantic buffer, smoothly transitioning from the original problem description to the appended keywords. The enriched problem is then input into the LLM to generate the final solution.

## 3 EXPERIMENTAL SETUP

We conduct a series of experiments to evaluate the effectiveness of the proposed approach SEK. In this section, we describe our experimental setup, including the selected models, benchmarks, evaluation metrics, baselines, and implementation details.

### 3.1 STUDIED LLMS

We select five representative LLMs to evaluate SEK, balancing between open-source and proprietary models, as well as covering a range of model sizes and architectures. The open-source models include Llama-3.1-70B-instruct (Dubey & Abhinav Jauhri, 2024), which is a dense decoder-only model with 70-billion parameters, Mixtral-8×22B-instruct-v0.1 (Jiang et al., 2024), which is a sparse Mixture-of-Experts (MOE) model having 141-billion total parameters with 39B active, and DeepSeek-Coder-V2-236B-Instruct-0724 (Zhu et al., 2024), which is a sparse MOE model having 236B parameters with 21B active. We access DeepSeek-Coder via DeepSeek-AI's API. For proprietary models, we include GPT-3.5-turbo-0125 (OpenAI, 2022) and GPT-4o-mini (OpenAI, 2024), accessed via OpenAI's API. Detailed specifications for each model are provided in the Appendix B.

### 3.2 BENCHMARKS AND EVALUATION METRIC

Following previous work (Chen et al., 2023b; Dong et al., 2023; Zhong et al., 2024b; Jiang et al., 2023b), We conduct experiments on three public code generation benchmarks HumanEval(+) (Chen et al., 2021; Liu et al., 2024), MBPP(+) (Austin et al., 2021; Liu et al., 2024), and APPS (Hendrycks et al., 2021). Considering the high cost of evaluating the entire APPS test problems and following prior work (Olausson et al., 2023; Huang et al., 2024b; Le et al., 2024; Yang et al., 2023), we randomly select 300 problems from the APPS test set for evaluation[1]. To mitigate the uncertainty introduced by random sampling, we conduct multiple experiments with different sample seeds. More details are in Appendix E.3. For detailed descriptions of each benchmark, please refer to Appendix C. We evaluate model performance using the Pass@1 metric, which measures the ability to generate correct solutions in a single attempt. This also aligns with real-world scenarios where developers aim to produce accurate code on the first try.

---

[1]There are three different difficulty levels of problems in APPS, i.e., introductory, interview, and competition. Specifically, based on the frequency distribution of problems with different difficulty levels, we sample 60, 180, and 60 problems at the introductory, interview and competition levels, respectively. All tasks are listed in Appendix E.3.

## 3.3 BASELINES

- **Default LLM**: This approach is based on the EvalPlus framework (Liu et al., 2024), using problems from the benchmark as input to prompt LLMs for code generation.
- **Zero-Shot CoT (Chain-of-Thought)** (Kojima et al., 2022): This approach first prompts the LLM to "think step by step" for getting the intermediate reasoning steps and then concatenates the original problem description with the generated intermediate steps as input to get the code solution.
- **CoT** (Wei et al., 2022): This approach generates a series of reasoning steps during the solution-generation process for each problem. To ensure comparative fairness, both the CoT baseline and SEK employ an equal number of demonstrations.
- **One-Step CoT**: This approach first prompts the LLM to "Rephrase the problem description using precise language", and then uses this refined description to guide code generation. Both One-Step CoT and SEK employ an equal number of demonstrations.
- **SelfEvolve** (Jiang et al., 2023a): This approach first uses LLMs to generate problem-specific knowledge and produce initial code solutions based on such knowledge. Then, it iteratively refines code solutions with LLMs based on execution feedback. Notably, SelfEvolve uses different prompt templates for different benchmarks to extract knowledge. Since these prompt templates have been open-sourced, we consistently apply its two-stage prompts on HumanEval (see Appendix H) in our replication process. For a fair comparison, we remove the self-refinement module, and employ the same number of demonstrations as SEK.
- **Beam Search** (Wiseman & Rush, 2016): This approach employs distinct search beams and optimizes selection during the decoding process. Given that SEK requires LLMs to explore search space twice by modifying the LLM's search space through additional token insertion, we demonstrate its benefit by comparing it with performing two searches within the LLM's original search space,i.e., beam search with a beam size of 2. We also demonstrate that with similar computational costs, SEK consistently outperforms beam search (Appendix E.4).

## 3.4 IMPLEMENTATION DETAILS

**Prompt Design.** It's worth noting that the implementations except SelfEvolve are based on the EvalPlus framework. Specifically, the only difference between SEK and Default is the addition of keywords and explanations to the problem description. APPS contains problems in two formats: call-based format and standard input format. Following previous work (Olausson et al., 2023; Inala et al., 2022; Chen et al., 2023b), we employ a two-shot prompt to guide the LLM to generate appropriate solutions for different formats.

**Demonstration Selection Strategy.** Inspired by previous work (Wei et al., 2022; Mu et al., 2023; Wang et al., 2023), we adopt a differentiated strategy that varies based on benchmark complexity (See Appendix D). To reduce bias, we employ an LLM separate from our target LLMs (Claude-3.5-Sonnet) to generate keywords and explanations for each demonstration, which are then manually reviewed and refined (See Appendix D).

**Configuration.** In our experiments, we treat the LLMs as black-box generators and only need to set a few key interface parameters. We maintain consistent settings across all LLMs, employing greedy decoding for output generation. The maximum output length is uniformly set to 2048 tokens. Specifically, the LLMs accessed via APIs do not support Beam Search. Thus, we only implement Beam Search for Llama-3.1-70B-Instruct and Mixtral-8×22B-Instruct-v0.1. Due to resource limitation, we compare SelfEvolve using GPT-3.5-turbo following the original paper (Jiang et al., 2023a) and additionally use two open-sourced LLMs (Llama-3.1 and Mixtral-8x22B).

## 4 EXPERIMENTAL RESULTS

## 4.1 MAIN RESULTS

Table 1 presents the performance of SEK and the selected baselines across five representative LLMs on Humaneval(+), MBPP(+) and APPS of different difficulty levels. To be noted, the Default results of Mixtral-8×22B-Instruct-v0.1 and DeepSeekCoder-V2-Instruct on Humaneval(+) and MBPP(+) are from the official leaderboard of the EvalPlus (Liu et al., 2024). However, as the other three LLMs are not in this leaderboard, we adhere to the EvalPlus framework to obtain their results.

| Model | Method | HumanEval | HumanEval+ | MBPP | MBPP+ | APPS Introductory | APPS Interview | APPS Competition | Average |
|---|---|---|---|---|---|---|---|---|---|
| Llama-3.1-70B-Instruct | Default | 78.0 | 73.8 | 87.6 | 70.9 | 50.0 | 15.0 | 5.0 | 54.3 |
| | Beam Search (2) | 79.3 | 74.4 | 87.8 | 70.9 | 55.0 | 16.1 | 5.0 | 55.5 |
| | One-Step CoT | 79.3 | 73.2 | 71.7 | 57.4 | 50.0 | 17.2 | 3.3 | 50.3 |
| | Zero-Shot CoT | 76.8 | 72.6 | 77.5 | 62,4 | 41.6 | 16.1 | **8.3** | 48.8 |
| | CoT | 79.9 | 74.4 | 87.0 | **71.7** | 43.3 | 16.6 | 6.7 | 54.2 |
| | SelfEvolve | 81.7 | 75.6 | 85.4 | 70.4 | 50.0 | 15.5 | **8.3** | 55.3 |
| | SEK | **84.8** | **79.3** | **88.4** | 71.2 | **61.7** | **20.0** | 8.3 | **59.1** |
| Mixtral-8×22B-Instruct-v0.1 | Default | 76.2 | 72.0 | 73.8 | 64.3 | 28.3 | 7.7 | 1.6 | 46.3 |
| | Beam Search (2) | 78.7 | 73.2 | **81.2** | **70.6** | **33.3** | 8.8 | **6.6** | 50.3 |
| | One-Step CoT | 72.0 | 66.5 | 79.6 | 66.9 | 31.6 | 6.1 | 1.6 | 46.3 |
| | Zero-Shot CoT | 75.0 | 68.3 | 79.9 | 67.2 | 28.3 | 8.3 | 1.6 | 46.9 |
| | CoT | 72.0 | 65.9 | 78.0 | 68.0 | 31.6 | 3.8 | 5.0 | 46.3 |
| | SelfEvolve | 56.7 | 50.0 | 68.5 | 60.1 | **33.3** | 7.2 | 5.0 | 40.1 |
| | SEK | **81.1** | **75.6** | 79.1 | 66.9 | **33.3** | **10.0** | 6.6 | **50.4** |
| GPT-3.5-turbo (API) | Default | 72.6 | 67.7 | **84.1** | 71.2 | 46.6 | 18.3 | 0.0 | 51.5 |
| | One-Step CoT | 70.1 | 65.9 | 78.6 | 66.1 | **53.3** | 16.1 | 1.6 | 50.2 |
| | Zero-Shot CoT | 72.6 | 67.1 | 83.3 | 71.2 | 48.3 | **20.6** | 3.3 | 52.3 |
| | CoT | 58.5 | 54.9 | **84.1** | 68.8 | 41.6 | 17.2 | 1.6 | 46.7 |
| | SelfEvolve | 73.2 | 67.7 | 82.3 | 66.7 | 45.0 | 19.4 | 1.6 | 50.8 |
| | SEK | **75.6** | **69.5** | **84.1** | 72.5 | **53.3** | **20.6** | 5.0 | **54.4** |
| GPT-4o-mini (API) | Default | **87.8** | 84.1 | 85.7 | 72.8 | 53.3 | 31.6 | 11.6 | 61.0 |
| | One-Step CoT | 86.0 | 79.3 | 85.4 | 70.9 | 45.0 | 29.4 | 10.0 | 58.0 |
| | Zero-Shot CoT | 86.6 | **84.8** | **89.7** | **76.2** | 33.3 | 27.2 | 8.3 | 58.0 |
| | CoT | 87.2 | 84.1 | 88.1 | 73.3 | 50.0 | 33.8 | 11.6 | 61.2 |
| | SEK | 87.2 | 84.1 | 87.8 | 74.1 | **58.3** | **35.0** | **13.3** | **62.8** |
| DeepSeekCoder-V2-Instruct (API) | Default | 85.4 | 82.3 | 89.4 | 75.1 | 70.0 | 36.1 | 10.0 | 64.0 |
| | CoT | 88.4 | 82.3 | **90.5** | 75.4 | 60.0 | 40.5 | 10.0 | 63.9 |
| | SEK | **93.3** | **85.4** | 90.2 | **76.2** | **75.0** | **41.1** | **13.3** | **67.8** |

Table 1: Pass@1 (%) results of SEK and baseline methods on HumanEval(+), MBPP(+) and APPS of different difficulty levels. **Bold** numbers indicate the best-performing baseline for each model.

Overall, SEK substantially improves code generation performance, achieving notable gains across various LLMs and datasets. We observe that SEK achieves greater performance improvements on HumanEval(+) and APPS than MBPP(+). For instance, on HumanEval, SEK demonstrates an absolute average performance improvement of 4.4% over the Default, whereas, it achieves an improvement of 1.8% on MBPP. This may be because the problems in HumanEval(+) and APPS are more complex than those in MBPP, and simple problems are easy to understand and alleviate the need to extract and explain keywords. As shown in Table 3, the average number of tokens per problem is 26.1 for MBPP, while those numbers are 67.7 and more than 257.3 for HumanEval(+) and APPS. These results may indicate that SEK can better improve LLMs' problem-solving capabilities on relatively complex problems than on simple problems.

We first discuss the performance on HumanEval(+) and APPS. These benchmarks are relatively complex compared to MBPP, and better demonstrate the effectiveness of SEK. SEK consistently outperforms Default across most LLMs. For instance, SEK achieves average absolute improvements of 6.7%, 3.6%, and 3.7% on APPS-Introductory, APPS-Interview, and APPS-Competition, respectively. However, GPT-4o-mini is an exception, which experiences a slight performance decline on Humaneval(+). This may be because the built-in prudence of GPT-4o-mini (Huang et al., 2024a) makes it tend to select more generic keywords, and such generic keywords fail to help LLMs understand low-frequency terms in the problem description. This conjecture is further underpinned by an observation that CoT similarly fails to enhance GPT-4o-mini's performance. The consistent improvements of SEK across most LLMs highlight its effectiveness in enhancing the problem-solving capabilities of LLMs.

Compared to Beam Search, which also explores the search space twice, SEK shows notable performance improvements. For instance, on Humaneval and Humaneval+, SEK achieves absolute average improvements of 4.0% and 3.7%, respectively, over Beam Search. These can be attributed to SEK's unique technique: appending the problem's critical parts to the end, enabling LLMs to focus on and comprehend these key concepts. In contrast, Beam Search merely expands the search space without understanding the problem deeply, leading to lower diversity in outputs (Li & Jurafsky, 2016). Consequently, it cannot enhance problem-solving capabilities in a targeted manner like SEK (See Appendix I for different cases).

Compared to CoT, SelfEvolve, One-Step CoT, and Zero-Shot CoT, SEK demonstrates a notable and consistent performance advantage. For instance, on Humaneval and Humaneval+, SEK achieves

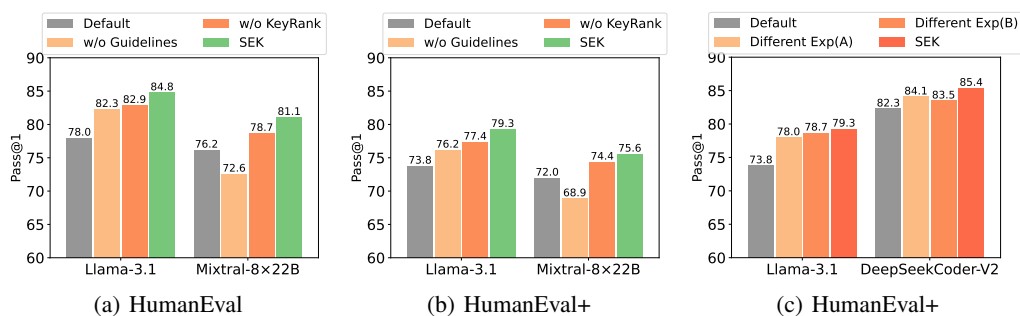

(a) HumanEval      (b) HumanEval+      (c) HumanEval+

Figure 4: (a-b) Ablation experiments on the Humaneval(+) benchmarks with two LLMs. (c) Different explanations of Demonstrations on Humaneval+ with two LLMs.

| Model | Method | Humaneval | Humaneval+ |
|---|---|---|---|
| Llama-3.1-70B-Instruct | Default | 78.0 | 73.8 |
| | SEK (corpus = APPS training set) | 84.1 | 78.7 |
| | SEK (corpus = Python subset of eval-codealpaca-v1) | **84.8** | **79.3** |
| DeepSeekCoder-V2-Instruct | Default | 85.4 | 82.3 |
| | SEK (corpus = APPS training set) | 90.9 | **85.4** |
| | SEK (corpus = Python subset of eval-codealpaca-v1) | **93.3** | **85.4** |

Table 2: SEK works under different corpus for Humaneval(+).

absolute average performance improvements of 7.2% and 6.5% over CoT. In contrast, the performance of the four baselines is inconsistent, sometimes even lower than Default. For instance, with Mixtral-8×22B-Instruct-v0.1, SelfEvolve's performance on APPS-Interview is 0.5% lower than Default. The unstable performance of CoT can be attributed to its inherent unsuitability for generation tasks (Sprague et al., 2024). Similar phenomena have been observed in prior work (Wang et al., 2024; Zhang et al., 2024; Luo et al., 2024; Jiang et al., 2023b). While the four baselines utilize LLMs to extract relevant knowledge from problem descriptions, they differ in the types of extracted knowledge. SEK focuses on low-frequency keywords, which are more difficult to be mapped to code implementation. This enables SEK to effectively fill the knowledge gaps during code generation. In contrast, the other three methods tend to merely restate the complete problem description for problems in code generation benchmarks. In addition, upon manual inspection of the generated problem descriptions for One-Step CoT, we identify that LLMs, without human intervention, often struggle to consistently produce precise whole-problem reformulations. Any errors in this intermediate generation step can compromise the overall description accuracy. In contrast, SEK focuses on analyzing specific keywords within the problem description, which helps mitigate the potential errors that might arise from whole-problem reformulation. As a result, the four baselines are less effective compared to SEK in code generation.

We then discuss the performance on MBPP(+), a relatively simple benchmark. SEK surpasses the baselines across most LLMs, further demonstrating SEK's effectiveness. For instance, when applied to Llama-3.1-70B-Instruct, SEK achieves performance improvements of 3.0% and 0.8% over SelfEvolve on MBPP and MBPP+, respectively.

## 4.2 DISCUSSION

We conduct additional experiments to comprehensively evaluate SEK's performance and robustness.

**Guidelines in the prompt for KeyExtract & Explain provide essential guidance for LLMs, KeyRank effectively prioritizes keywords, and generated explanations are important**. Our ablation studies confirm that both guidelines and KeyRank play crucial roles in enhancing performance. As shown in Figure 4(a)-4(b), We evaluate Llama-3.1 and Mixtral-8×22B on Humaneval (+). Removing either the guidelines or the KeyRank module results in performance degradation. For instance, removing the KeyRank module results in performance decreases of 2.4% and 1.2% on HumanEval and HumanEval+, respectively, for Mixtral-8×22B-Instruct-v0.1. Moreover, removing each guideline from the prompt individually also results in performance degradation in most cases

Figure 5: A real case from MBPP generated by two baselines and SEK.

(See Appendix E.2). It is worth mentioning that even without KeyRank, SEK remains superior to the Default baseline. For instance, without KeyRank module, Mixtral-8×22B-Instruct-v0.1 shows a 2.5% improvement on HumanEval compared to the Default, underscoring the strength of SEK's core mechanisms. We also conduct an ablation study by removing generated explanations from the enriched prompts. Experimental results show that removing these explanations leads to substantial performance drops across different LLMs, demonstrating the importance of generated explanations. See Appendix E.6 for more details.

**SEK demonstrates robustness to variations in demonstrations, and the corpus used in KeyRank**. To show its performance is not tied to a fixed set of keyword explanations within the demonstrations used in KeyExtract & Explain, We conduct experiments using two additional sets of keyword explanations randomly generated from the same LLM (i.e., Claude-3.5-Sonnet). As shown in Figure 4(c), although there is performance variance among different keyword explanations, as would be expected when using exemplar-based prompting (Gao et al., 2021; Min et al., 2022; Reynolds & McDonell, 2021), the three sets of keyword explanations consistently outperform the Default. Additionally, to evaluate the robustness to the corpus used in KeyRank, we employ select different corpus, as shown in Table 2. We observe that using SEK with Llama-3.1-70B-Instruct still shows a 6.1% absolute improvement on Humaneval compared to Default. These results demonstrate the robustness of SEK.

**SEK enhances the model's focus on core keywords in the problem description** (See Appendix G). Using a visualization tool, we analyze SEK's behaviors from the perspective of attention distribution. We select a simple problem, i.e., "Write a function to find the nth nonagonal number", choosing the keyword "nonagonal" with its explanation for detailed analysis. By comparing the attention distribution in the Default and SEK settings, we observe that SEK help the LLM allocate more attention to the keyword and its explanation. This indicates the way SEK uses to enrich the prompt can help LLMs better focus on the key concepts in the problem description, leading to improved code generation.

## 4.3 CASE STUDY

To further evaluate the effectiveness of SEK, we conduct a qualitative analysis. As shown in Figures 5, we select one representative sample from MBPP, use DeepSeek-Coder-V2-Instruct as the base model, and compare the outputs of SEK with Default and CoT. See Appendix J for more cases.

The problem aims to find the kth element in the given array using 1-based indexing. The solutions generated by Default and CoT both perform unnecessary sorting and are incorrect. This may be because the LLM incorrectly correlates the keyword *kth element* with the sorting operation. In contrast, SEK accurately interprets *kth element* and produces the correct code solution. This is achieved by incorporating the guideline that ensures the explanations are consistent with test cases in the problem description, demonstrating the effectiveness of SEK.

## 5 RELATED WORK

**LLM-based code generation**: Recent advancements in LLMs have significantly improved code generation capabilities. Models like CodeGen (Nijkamp et al., 2022), StarCoder (Li et al., 2023), and GPT series (Black et al., 2022; Chen et al., 2021) have demonstrated remarkable performance in translating natural language descriptions into code snippets. These models primarily use decoder-only architectures and next-token prediction for pre-training. A subset, including CodeT5 (Wang et al., 2021) and PLBART (Ahmad et al., 2021), employs encoder-decoder architectures. Our work builds upon these foundations, focusing on enhancing LLMs' problem-solving capabilities without additional training.

**Prompting techniques for code generation**: Prompting techniques for code generation can be broadly categorized into three types: The first type utilizes external knowledge to enhance LLMs' understanding of coding problems or intermediate outputs (Mu et al., 2023; Nashid et al., 2023; Zhong et al., 2024a). For example, CEDAR (Nashid et al., 2023) retrieves relevant code examples from an external knowledge base to help LLMs understand task requirements. The second type relies solely on LLMs' inherent capabilities, using prompt design to guide LLMs in generating code snippets that meet specific requirements (Wei et al., 2022; Wang et al., 2023; Yao et al., 2024). For instance, Chain of Thought (Wei et al., 2022) employs a step-by-step, chain-of-thought style prompt to guide LLMs in producing correct results. The third type integrates the previous two types, leveraging both external knowledge and the LLM's inherent knowledge to solve coding problems (Chen et al., 2023c; Jiang et al., 2023a; Tian & Chen, 2023; Chen et al., 2024c;b). For example, Self-Debug (Chen et al., 2023c) uses the code execution results or the code explanations generated by the LLM itself to debug the incorrect code multiple times. SEK belongs to the second category. Different from other methods, it focuses on improving LLMs' comprehension of the problem by identifying and explaining the key concepts in the problem description with LLMs themselves.

**Keyword extraction**: Keyword extraction methods have evolved from traditional statistical (Sparck Jones, 1972; El-Beltagy & Rafea, 2009; Florescu & Caragea, 2017; Rose et al., 2010) and graph-based approaches (Mihalcea & Tarau, 2004; Wan & Xiao, 2008; Gollapalli & Caragea, 2014; Grineva et al., 2009) to more advanced techniques leveraging language models (Mahata et al., 2018; Bennani-Smires et al., 2018; Sun et al., 2020; Arora et al., 2017). Recent works like Attention-Rank (Ding & Luo, 2021) and LLM-TAKE (Maragheh et al., 2023) use self-attention mechanisms and language models to identify significant keywords. Our work extends this concept to the domain of code generation, using LLMs to extract and explain problem-specific keywords to enhance code solution generation.

## 6 CONCLUSION AND LIMITATIONS

In this work, we propose SEK, a simple yet effective method to enhance the code generation capabilities of LLMs. SEK leverages the LLM to extract and explain keywords from the problem description, followed by ranking them based on their frequency. Through extensive experiments, we demonstrate that SEK facilitates LLMs in capturing and clarifying key concepts within problems, thereby generating more accurate code solutions.

One limitation of SEK is that the two-stage invocation process of SEK incurs additional computational overhead. Future work could explore compressing the process into one invocation. In addition, keywords are extracted and explained by LLMs, of which the quality cannot be guaranteed due to the hallucinations of LLMs (Ji et al., 2023). Mitigating this requires enhancing the factual accuracy of LLMs (Mitchell et al., 2022; Tang et al., 2023) and proposing effective approaches for detecting factual errors (Chen et al., 2024a; Min et al., 2023).

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

# A AGLORITHM OF KEYRANK

---

**Algorithm 1** KeyRank Procedure

---

**Input:** Keyword Set $K_x$, Problem $\mathbf{P}$, Corpus $C$
**Output:** Ranked Keywords $K_y$
1: $K_g \leftarrow \emptyset$, $K_a \leftarrow \emptyset$, $K_y \leftarrow \emptyset$
2: $\mathbf{f} \leftarrow$ EXTRACTFUNCTIONNAME($\mathbf{P}$)
3: **for** each $k$ in $K_x$ **do**
4:     **if** $k = \mathbf{f}$ **then**
5:         $K_g \leftarrow K_g \cup \{(k, -1)\}$
6:     **else if** $k \in \mathbf{P}$ **then**
7:         $K_g \leftarrow K_g \cup \{(k, \text{TF-IDF}(k, \mathbf{P}, C))\}$
8:     **else**
9:         $K_a \leftarrow K_a \cup \{k\}$
10:    **end if**
11: **end for**
12: $K_g \leftarrow$ SORTDESCENDING($K_g$)
13: $K_y \leftarrow K_a \cup K_g$
14: **return** $K_y$

---

First, we initialize the *General Keywords*, *Abstract Keywords*, and output as $K_g, K_a, K_y$, respectively. EXTRACTFUNCTIONNAME extracts the method name if provided in the problem description. Otherwise, it returns a null value. Then, keywords are classified and scored. They can be divided into three classes: *Abstract Keywords*, *General Keywords*, and *Function Keyword*. Abstract keywords do not appear in any input; they are abstract terms summarized from multiple concepts and stored in $K_a$. General keywords denote items in the problem description. We calculate their importance using TF-IDF based on a code-related corpus. General keywords and their scores are stored in $K_g$. Function keyword refers to the method name for solving the problem. Its explanation provides a coarse-grained description of the problem requirements. We assign a score of -1 to the function keyword, and also store them in $K_g$. Finally, SORTDESCENDING sorts the keywords in $K_g$ based on their scores. The keywords are combined in the order of abstract, general, and function keywords, and are then returned as the Ranked Keywords.

# B STUDIED LLMS

- **Llama-3.1-70B** (Dubey & Abhinav Jauhri, 2024) is an open-sourced, decoder-only language model, pre-trained on 15t tokens from public sources. In our experiments, we use the Llama-3.1-70B-Instruct version.
- **Mixtral-8×22B** (Jiang et al., 2024) is an open-source, sparse Mixture-of-Experts (MOE) model with 141B total parameters, utilizing 39B active parameters. We use the Mixtral-8×22B-Instruct-v0.1 version.
- **DeepSeek-Coder-V2-Instruct-0724** (Zhu et al., 2024), developed by DeepSeek-AI, is an open-source MoE code language model pre-trained on 10.2T tokens. The instruction-tuned version is further trained on 11B tokens.
- **GPT-3.5-turbo-0125** (OpenAI, 2022) is a close-sourced LLM from OpenAI, building on GPT-3 with optimizations for more efficient text generation.
- **GPT-4o-mini** (OpenAI, 2024) is a smaller, cost-effective[2] variant of GPT-4 (OpenAI & Josh Achiam, 2024), offering strong performance across various tasks.

---

[2]GPT-4 is not selected due to the high experimental cost required.

## C    BENCHMARK DETAILS

| Benchmark | Humaneval | Humaneval+ | MBPP | MBPP+ | APPS Introductory | APPS Interview | APPS Competition |
|---|---|---|---|---|---|---|---|
| Problem | 164 | 164 | 399 | 399 | 60 | 180 | 60 |
| #Avg Tests | 9.6 | 764.1 | 3.1 | 105.4 | 15.1 | 25.7 | 17.3 |
| #Avg Tokens | 67.7 | 67.7 | 26.1 | 26.1 | 257.3 | 319.8 | 377.4 |

Table 3: Statistics of benchmarks: the total number of problems in each benchmark (Problems), the average number of hidden test cases per problem (#Avg Tests), and the average number of space-separated tokens of the problem (#Avg Tokens).

We use three widely-used benchmarks, i.e., HumanEval(+), MBPP(+), and APPS, for evaluation. Table 3 presents their key statistics.
(1) **HumanEval** (Chen et al., 2021) consists of 164 hand-written programming problems, each including a method signature, docstring, body, and unit tests. We use both HumanEval and its extended version, HumanEval+(Liu et al., 2024), which enhances the original with $80\times$ additional test samples to address test case insufficiency (Liu et al., 2024).
(2) **MBPP** (Austin et al., 2021) contains crowd-sourced Python programming problems. Our study uses the versions proposed by (Liu et al., 2024), including MBPP and MBPP+. Each of them contain 399 tasks, and the latter adds $35\times$ test samples.
(3) **APPS** (Hendrycks et al., 2021) includes 10,000 coding problems from open-access websites, split equally into training and test sets. It includes two problem formats: call-based format (input via function parameters) and standard input format (using stdin/stdout). Problems are categorized into introductory, interview, and competition levels. There are three different difficulty levels of problems in APPS, i.e., introductory, interview and competition. Each of them has 1000, 3000, and 1000 tasks, respectively. Considering the cost of evaluating the entire APPS test set and following prior work (Olausson et al., 2023; Huang et al., 2024b; Le et al., 2024; Yang et al., 2023), we randomly select problems in accordance with the frequency distribution of these difficulty levels and sample 60, 180, 60 problems at the introductory, interview, and competition levels, respectively.

## D    IMPLEMENTATION DETAILS

**Demonstration selection strategy.** Specifically, for HumanEval, we select the first two problems as demonstrations. For MBPP, we choose the first problem. For APPS, considering the model's input length limitation and to avoid randomness, we select the two shortest problems from the first five problems in the training set. The reason for this differentiated strategy is that HumanEval and APPS problems are more complex, requiring more examples, while MBPP problems are relatively simple in form, and one example is enough.

**Keywords and explanations involved in demonstrations.** The prompt for KeyExtract & Explain uses several demonstrations to guide LLMs to produce keywords and their explanations. To ensure the quality of each demonstration, we first employ Claude-3.5-Sonnet, an LLM separate from our target LLMs, to generate multiple sets of keywords and explanations for each demonstration. The generated contents are then manually reviewed, and the most accurate set for each demonstration is selected and used in the prompt. This can mitigate the potential bias in human-generated explanations. Additionally, for HumanEval(+) and MBPP(+) datasets, which provide function names, the first two authors discuss and write the explanation for the function name in each demonstration.

## E    ADDITIONAL EXPERIMENTS

### E.1    INFLUENCE OF KEYWORD COMBINATION ORDERS

In KeyRank, we combine different types of keywords based on the order of *abstract → general → function*. We investigate the influence of keyword combination orders by comparing the order used by SEK with three alternative ordering strategies using two LLMs, i.e., Llama-3.1-70B-Instruct

| Model | Combination Order | HumanEval | HumanEval+ | Average |
|---|---|---|---|---|
| | Default | 78.0 | 73.8 | 75.9 |
| | Func_Abs_Gen | 83.5 | 78.7 | 81.1 |
| | Func_Gen_Abs | 84.1 | **79.3** | 81.7 |
| Llama-3.1-70B-Instruct | Gen_Func_Abs | 84.1 | 78.7 | 81.4 |
| | Gen_Abs_Func | 84.1 | 78.7 | 81.4 |
| | Abs_Func_Gen | 84.1 | 78.0 | 81.1 |
| | SEK(Abs_Gen_Func) | **84.8** | **79.3** | **82.1** |
| | Default | 76.2 | 72.0 | 74.1 |
| | Func_Abs_Gen | 78.0 | 72.0 | 75.0 |
| | Func_Gen_Abs | **81.1** | 75.0 | 78.1 |
| Mixtral-8×22B-Instruct-v0.1 | Gen_Func_Abs | 78.0 | 72.0 | 75.0 |
| | Gen_Abs_Func | 76.8 | 71.3 | 74.1 |
| | Abs_Func_Gen | **81.1** | **75.6** | **78.4** |
| | SEK(Abs_Gen_Func) | **81.1** | **75.6** | **78.4** |

Table 4: The experiments of different combination orders on Humaneval(+) with two LLMs.

| Model | Ablations | HumanEval | HumanEval+ |
|---|---|---|---|
| | w/o Guideline(1) | **85.4** | 78.7 |
| | w/o Guideline(2) | 82.3 | 75.6 |
| | w/o Guideline(3) | 81.7 | 76.8 |
| Llama-3.1-70B-Instruct | w/o Guideline(4) | 81.1 | 76.2 |
| | w/o Guideline(5) | 83.5 | 77.4 |
| | ALL Guidelines | 84.8 | **79.3** |
| | w/o Guideline(1) | 76.8 | 72 |
| | w/o Guideline(2) | 77.4 | 72.6 |
| | w/o Guideline(3) | 79.3 | 73.8 |
| Mixtral-8×22B-Instruct-v0.1 | w/o Guideline(4) | 75.0 | 70.1 |
| | w/o Guideline(5) | 76.8 | 73.2 |
| | ALL Guidelines | **81.1** | **75.6** |

Table 5: Ablation experiments on removing one guideline at a time from Keyword Prompt on HumanEval(+) with two LLMs.

and Mixtral-8×22B-Instruct-v0.1. Table 4 presents the experimental results, where the abbreviations Abs, Gen, and Func denote *abstract keywords*, *general keywords*, and *function keywords*, respectively. The results reveal performance variations across different keyword combination orders, indicating that the order of different keyword types impacts LLMs' comprehension of coding problems. The combination order used by SEK consistently yields optimal performance, suggesting its rationality.

### E.2 INFLUENCE OF GUIDELINES

In Section 4.2, we investigate the effectiveness of the guidelines in the KeyExtract & Explain prompt as a whole. This section further investigates the impact of each guideline by removing it from the prompt and re-evaluate the performance of SEK with two LLMs, i.e., Llama-3.1-70B-Instruct and Mixtral-8×22B-Instruct-v0.1 on HumanEval(+). Table 5 presents the experimental results, where the performance of the two LLMs decreases in almost all cases, indicating the contribution of each guideline to the effectiveness of SEK.

### E.3 MORE EXPERIMENTS ON APPS

In the main experiment, we randomly sample problems from the APPS test set for evaluation due to limited resources. The performance of LLMs on APPS may be affected by the randomness of the selected samples. To mitigate this variability, we conduct additional experiments by randomly selecting three new subsets of problems at the introductory level from the APPS test set and using two LLMs for evaluation, i.e., Llama-3.1-70B-instruct and GPT-3.5-Turbo. The number of sampled tasks is fixed at 60, consistent with the main experiment. For reproducibility, the selected tasks

| Model | Method | Introductory(A) | Introductory(B) | Introductory(C) | Average |
|---|---|---|---|---|---|
| | Default | 51.6 | 45.0 | 46.6 | 47.7 |
| | Beam Search(2) | 55.0 | 45.0 | 45.0 | 48.3 |
| | One-Step CoT | 48.3 | 48.3 | 48.3 | 48.3 |
| Llama-3.1-70B-Instruct | Zero-Shot CoT | 41.6 | 40.0 | 30.0 | 37.2 |
| | CoT | 41.6 | 46.6 | 45.0 | 44.4 |
| | SelfEvolve | 45.0 | 53.3 | 46.6 | 48.3 |
| | **SEK** | **58.3** | **56.6** | **50.0** | **55.0** |
| | Default | 45.0 | 51.6 | 43.3 | 46.6 |
| | One-Step CoT | **53.3** | 48.3 | 41.6 | 47.7 |
| GPT-3.5-turbo | Zero-Shot CoT | 48.3 | 51.6 | **50.0** | 50.0 |
| (API) | CoT | 48.3 | **53.3** | 46.6 | 49.4 |
| | SelfEvolve | 45.0 | 48.3 | 45.0 | 46.1 |
| | **SEK** | 48.3 | **53.3** | **50.0** | **50.5** |

Table 6: The Pass@1 (%) results of SEK and baseline methods on differently sampled APPS-Introductory sets.

are provided in Table 11. As shown in Table 6, SEK achieve optimal performance across different subsets. For instance, considering Llama-3.1-70B-Instruct, SEK outperforms the Default, Beam Search, and CoT baselines by an average of 7.3%, 6.7%, and 10.6%, respectively. This corroborates the credibility of our conclusions.

### E.4 ANALYSIS OF PERFORMANCE AND COMPUTATIONAL COSTS OF BEAM SEARCH AND SEK

| Method | HumanEval | HumanEval+ | MBPP | MBPP+ | APPS Introductory | APPS Interview | APPS Competition | Average |
|---|---|---|---|---|---|---|---|---|
| Default | 78.0 | 73.8 | 87.6 | 70.9 | 50.0 | 15.0 | 5.0 | 54.3 |
| Beam Search(2) | 79.3 | 74.4 | 87.8 | 70.9 | 55.0 | 16.1 | 5.0 | 55.5 |
| Beam Search(3) | 78.0 | 74.4 | 87.8 | 72.2 | 53.3 | 20.0 | 6.6 | 56.0 |
| Beam Search(5) | 79.9 | 75.6 | 88.4 | **72.8** | 55.0 | **21.1** | 6.7 | 57.1 |
| Beam Search(10) | 79.9 | 75.0 | **88.9** | 72.5 | 56.6 | **21.1** | **8.3** | 57.5 |
| SEK | **84.8** | **79.3** | 88.4 | 71.2 | **61.7** | 20.0 | **8.3** | **59.1** |

Table 7: The Pass@1 (%) results of Llama-3.1-Instruct-70B of SEK and different number of beam sizes of beam search baselines on HumanEval(+), MBPP(+) and APPS of different difficulty levels.

| Method | Introductory(A) | Introductory(B) | Introductory(C) | Average |
|---|---|---|---|---|
| Default | 51.6 | 45.0 | 46.6 | 47.7 |
| Beam Search(2) | 55.0 | 45.0 | 45.0 | 48.3 |
| Beam Search(3) | 50.0 | 45.0 | 45.0 | 46.7 |
| Beam Search(5) | 53.3 | 43.3 | 43.3 | 46.6 |
| Beam Search(10) | 53.3 | 45.0 | 48.3 | 48.9 |
| **SEK** | **58.3** | **56.6** | **50.0** | **55.0** |

Table 8: The Pass@1 (%) results of Llama-3.1-Instruct-70B of SEK and different number of beam sizes of beam search baselines on differently sampled APPS-Introductory sets.

To investigate the impact of beam size on performance, we conduct additional experiments with varying beam sizes (2, 3, 5, and 10) using LLaMA-3.1-Instruct-70B. We are unable to include Mixtral-8×22B-Instruct-v0.1 in these experiments due to memory constraints (Out-Of-Memory issues) at beam sizes ≥ 5. The results, presented in Table 7 and Table 8, demonstrate that SEK consistently outperforms beam search across most scenarios, even with larger beam sizes. Interestingly, we observed that beam sizes of 5 and 10 occasionally surpassed SEK's performance on MBPP(+) and APPS-Interview, which may be attributed to more computation cost of beam search (see below for details).

| Method | HumanEval | MBPP | APPS Introductory | APPS Interview | APPS Competition | Average |
|---|---|---|---|---|---|---|
| Beam Search(2) | 242.0 | 378.0 | 202.0 | 304.0 | 416.0 | 308.4 |
| Beam Search(3) | 723.0 | 538.0 | 286.0 | 435.0 | 611.0 | 518.6 |
| Beam Search(5) | 1200.0 | 890.0 | 455.0 | 685.0 | 1165.0 | 879.0 |
| Beam Search(10) | 2500.0 | 1840.0 | 960.0 | 1360.0 | 2410.0 | 1814.0 |
| SEK | 450.0 | 412.0 | 273.0 | 337.0 | 484.0 | 391.2 |

Table 9: The computational resource usage of SEK and Beam search with different beam sizes. Underline number means the closest computational resource consumption to that of SEK of the same benchmark.

| Method | Introductory(A) | Introductory(B) | Introductory(C) | Average |
|---|---|---|---|---|
| Beam Search(2) | 192.0 | 200.0 | 202.0 | 198.0 |
| Beam Search(3) | 281.6 | 308.0 | 308.0 | 299.2 |
| Beam Search(5) | 460.0 | 485.0 | 480.0 | 475.0 |
| Beam Search(10) | 970.0 | 1050.0 | 950.0 | 990.0 |
| SEK | 270.0 | 269.0 | 281.0 | 273.3 |

Table 10: The computational resource usage of SEK and Beam search with different beam sizes on differently sampled APPS-Introductory sets. Underline number means the closest computational resource consumption to that of SEK of the same benchmark.

To quantify the computational resource usage of each approach, we calculated the product of the numbers of generated tokens and maintained paths as the total computational cost. The computational cost are shown in Tables 9 and Table 10. When comparing the scenarios with similar computational costs, SEK consistently outperforms beam search. In the cases where beam search surpasses SEK, beam search typically demands significantly more computational resources. For instance, on MBPP, beam search with sizes 5 and 10 consumed approximately 890 and 1840 computational units respectively, whereas SEK required only 412 units. These results reinforce SEK's efficiency in achieving superior performance.

### E.5 FREQUENCY OF EXTRACTED KEYWORDS

To validate whether the keywords extracted in the KeyExtract & Explain phase are relatively low-frequency terms, we conduct a comparative analysis between extracted keywords and other terms in problem descriptions. Specifically, we choose the keywords generated by Llama-3.1-70B-Instruct on HumanEval for analysis and use a controlled comparison where the extracted keywords are compared with other terms of the same n-gram length. We use TF-IDF scores as a proxy to assess the frequency of the terms. We conduct three separate experiments with different instruction tuning datasets and pertaining datasets for IDF calculations, including eval-codealpaca-v1 (Luo et al., 2023), OSS-Instruct (Wei et al., 2024) and randomly selected samples from Python subset of the Stack-V2 (Lozhkov et al., 2024), which is pre-training data of the StarCoder2.

As shown in Figure 6(a), Figure 6(b), and Figure 6(c), all experiments demonstrate consistent results: the distribution of extracted keywords exhibits a notable right-skewed pattern compared to other terms, indicating higher TF-IDF scores. This dual empirical analysis provides supporting evidence that SEK tends to identify relatively low-frequency terms as keywords.

### E.6 IMPORTANCE OF GENERATED EXPLANATIONS

To validate the effectiveness of keyword explanations generated in the KeyExtract & Explain step, we conduct an additional ablation experiment by removing the generated explanations while retaining the extracted keywords for code generation. We follow the same experimental setup on HumanEval(+) using Llama-3.1-70B-Instruct and GPT-3.5-turbo. The results are shown in Table 12. It can be observed that removing generated explanations from the enriched prompts leads to performance drops, demonstrating the importance of these explanations for the code generation process.

| Datasets | Tasks |
|---|---|
| Introductory(A) | 4029, 4032, 4050, 4054, 4060, 4099, 4116, 4131, 4132, 4148, 4157, 4166, 4180, 4206, 4211, 4232, 4251, 4256, 4283, 4289, 4317, 4320, 4323, 4332, 4343, 4356, 4417, 4451, 4469, 4471, 4527, 4538, 4541, 4542, 4546, 4599, 4625, 4640, 4676, 4680, 4704, 4721, 4748, 4774, 4780, 4781, 4787, 4800, 4806, 4826, 4837, 4864, 4868, 4878, 4888, 4896, 4924, 4926, 4930, 4943 |
| Introductory(B) | 4021, 4046, 4051, 4073, 4115, 4127, 4138, 4140, 4156, 4163, 4201, 4225, 4230, 4233, 4236, 4263, 4270, 4294, 4295, 4347, 4358, 4375, 4376, 4407, 4424, 4446, 4453, 4454, 4460, 4469, 4478, 4486, 4489, 4528, 4547, 4570, 4580, 4596, 4638, 4644, 4656, 4678, 4692, 4695, 4726, 4730, 4735, 4740, 4780, 4803, 4842, 4869, 4871, 4890, 4905, 4918, 4932, 4947, 4970, 4975 |
| Introductory(C) | 4018, 4028, 4042, 4085, 4102, 4134, 4146, 4172, 4201, 4203, 4257, 4274, 4281, 4309, 4314, 4324, 4353, 4356, 4387, 4418, 4447, 4461, 4465, 4473, 4474, 4486, 4499, 4506, 4510, 4528, 4560, 4592, 4604, 4617, 4627, 4701, 4704, 4710, 4712, 4713, 4716, 4719, 4723, 4732, 4763, 4772, 4801, 4812, 4867, 4879, 4893, 4898, 4907, 4924, 4929, 4944, 4957, 4973, 4974, 4975 |

Table 11: The tasks in different sampling Introductory sets.

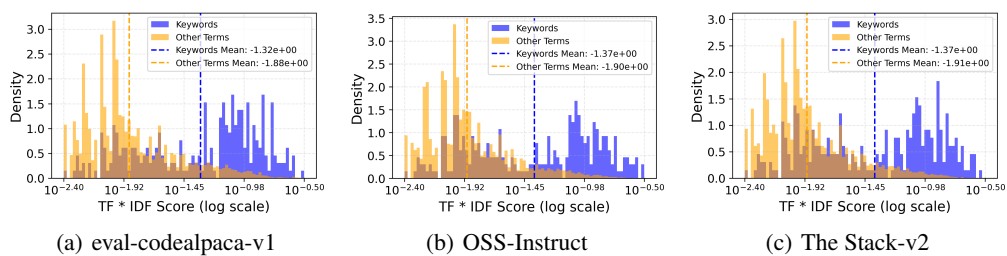

(a) eval-codealpaca-v1  (b) OSS-Instruct  (c) The Stack-v2

Figure 6: Comparison of the distribution of extracted Keywords and other terms with different corpus.

# F  SELECTED APPS TASKS

For reproducibility, we provide the complete list of selected tasks of APPS in Table 13.

# G  ATTENTION ANALYSIS

We aim to explain SEK from the perspective of attention distribution. We use BertViz[3] to present explainability visualizations. Due to limited computational resources, we select a short problem and remove its test cases. Specifically, the problem description is "Write a function to find the nth nonagonal number." and we select a keyword with its explanation "[nonagonal]: A nine-sided polygon. Nonagonal numbers represent the count of dots forming nonagons of increasing size". We select Mixtral-8×22B-Instruct-v0.1 as the base model and extract the attention from its last layer for analysis.

The key to this problem lies in understanding "nonagonal". With Default, Figure 7 shows the overall attention distribution for the problem, while Figure 8 displays the attention distribution for a part of the keyword "nonagonal". It can be observed that most of the attention is allocated to the beginning words, with the keyword "nonagonal" receiving relatively less attention. This may lead to insufficient focus on the core concept of the problem when generating code. In contrast, with SEK, Figure 10 presents the overall attention distribution of the LLM with SEK, and Figure 9 shows the attention distribution for "nonagonal". It can be seen that the model allocates additional attention

---

[3]https://github.com/jessevig/bertviz

| Model | Method | Humaneval | Humaneval+ |
|-------|--------|-----------|------------|
| Llama-3.1-70B-Instruct | Default | 78.0 | 73.8 |
| | SEK w/o explanations | 78.7 | 74.4 |
| | **SEK** | **84.8** | **79.3** |
| GPT-3.5-turbo (API) | Default | 72.6 | 67.7 |
| | SEK w/o explanations | 72.6 | 68.9 |
| | **SEK** | **75.6** | **69.5** |

Table 12: Ablation experiments on removing generated explanations on HumanEval(+) with two LLMs.

| Difficulty | Tasks |
|------------|-------|
| Introductory | 4007, 4032, 4049, 4050, 4054, 4060, 4114, 4116, 4132, 4148, 4157, 4166, 4180, 4211, 4215, 4232, 4251, 4283, 4289, 4317, 4323, 4332, 4343, 4356, 4372, 4417, 4439, 4451, 4469, 4527, 4540, 4541, 4546, 4549, 4582, 4585, 4599, 4625, 4631, 4640, 4676, 4678, 4704, 4721, 4774, 4781, 4787, 4800, 4806, 4826, 4837, 4861, 4864, 4868, 4878, 4888, 4924, 4926, 4929, 4930 |
| Interview | 6, 10, 35, 44, 56, 76, 82, 95, 105, 106, 115, 133, 135, 178, 188, 198, 210, 213, 231, 240, 248, 278, 300, 305, 319, 342, 357, 372, 377, 379, 420, 457, 460, 483, 484, 489, 546, 553, 566, 567, 584, 634, 664, 669, 675, 686, 696, 701, 734, 785, 817, 855, 861, 876, 903, 909, 914, 932, 973, 989, 993, 994, 1017, 1020, 1025, 1033, 1039, 1053, 1069, 1101, 1122, 1132, 1140, 1144, 1158, 1166, 1167, 1224, 1226, 1232, 1280, 1313, 1346, 1351, 1361, 1373, 1375, 1391, 1394, 1406, 1409, 1432, 1458, 1459, 1478, 1487, 1491, 1508, 1520, 1527, 1534, 1540, 1557, 1563, 1565, 1590, 1635, 1640, 1715, 1720, 1733, 1749, 1761, 1768, 1775, 1813, 1823, 1833, 1838, 1864, 1881, 1885, 1955, 1976, 1982, 1989, 2003, 2006, 2011, 2015, 2048, 2053, 2062, 2077, 2097, 2101, 2145, 2177, 2192, 2209, 2273, 2293, 2317, 2361, 2406, 2443, 2492, 2494, 2495, 2502, 2513, 2514, 2533, 2542, 2546, 2552, 2554, 2609, 2615, 2641, 2642, 2655, 2657, 2684, 2707, 2725, 2726, 2728, 2729, 2762, 2767, 2776, 2784, 2788, 2815, 2850, 2874, 2914, 2982, 2999 |
| Competition | 3009, 3024, 3031, 3071, 3097, 3131, 3138, 3171, 3188, 3204, 3206, 3210, 3211, 3252, 3262, 3263, 3298, 3301, 3313, 3319, 3326, 3372, 3379, 3445, 3456, 3479, 3481, 3501, 3517, 3535, 3573, 3579, 3618, 3629, 3654, 3680, 3684, 3690, 3713, 3721, 3727, 3731, 3733, 3745, 3762, 3775, 3786, 3788, 3802, 3803, 3843, 3863, 3882, 3886, 3893, 3901, 3943, 3945, 3948, 3972 |

Table 13: The tasks in different difficulty levels of APPS.

to the added keywords and explanations, encouraging the model to focus more on the core concepts of the problem. With SEK, the LLM further distributes attention to the added keywords and explanations, which can enhance its understanding of the key concepts in the problem.

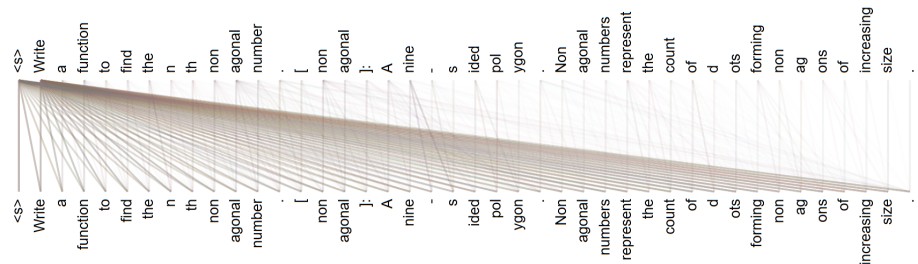

Figure 10: Final layer of the attention visualization with SEK

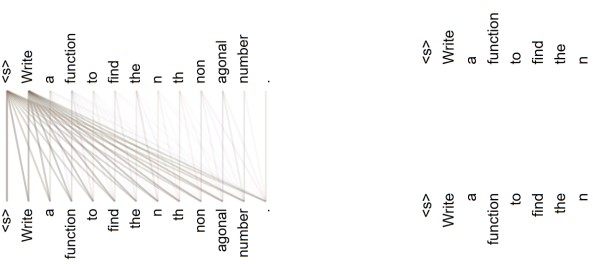

Figure 7: The overall attention visualization with Default.

Figure 8: Attention visualization for a part of the keyword "nonagonal" with Default.

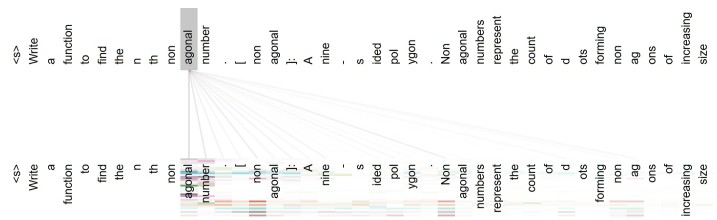

Figure 9: Attention visualization for a part of the keyword "nonagonal" with SEK.

## H  PROMPT FOR SELF-EVOLVE

```
```
{Problem description}
```
For the above question, could you briefly teach me how to solve it step by step in natural language? 'Dont
    write the code in this step.
```

Listing 1: The first prompt of Self-Evolve

```
Based on the above idea, help me complete the function.
Be attention, you should only output the codes without any explanation and natural language. Wrap your code
    with "```"
```

Listing 2: The Second prompt of Self-Evolve

## I  CASE STUDY OF THE DIFFERENCE BETWEEN BEAM SEARCH AND SEK

```
def digits(n):
    """Given a positive integer n, return the product of the odd digits.
    Return 0 if all digits are even.
    For example:
    digits(1)  == 1
    digits(4)  == 0
    digits(235) == 15
    """
```

Listing 3: A problem from Humaneval (problem ID 131)

```
def digits(n):
    """
    Given a positive integer n, return the product of the odd digits.
    Return 0 if all digits are even.

    Args:
        n (int): A positive integer.

    Returns:
        int: The product of the odd digits of n, or 0 if all digits are even.
    """
    product = 1
    for digit in str(n):
        digit = int(digit)
        if digit % 2 != 0:
            product *= digit
```

```
return product if product != 1 else 0
```

Listing 4: The output of Default from Humaneval (problem ID 131)

```
def digits(n):
    """Given a positive integer n, return the product of the odd digits.
    Return 0 if all digits are even.
    For example:
    digits(1)   == 1
    digits(4)   == 0
    digits(235) == 15
    """
    product = 1
    for digit in str(n):
        if int(digit) % 2 != 0:
            product *= int(digit)
    return product if product != 1 else 0
```

Listing 5: The output of Beam Search from Humaneval (problem ID 131)

```
[positive integer n]: The input parameter, a whole number greater than zero, which is the subject of the
    operation. The test cases demonstrate this concept with inputs 1, 4, and 235.
[product of the odd digits]: The result of multiplying all the odd digits together. In the test case
    digits(235), the product of the odd digits is 3 * 5 = 15.
[odd digits]: Refers to the individual numerical digits within the input integer n that are not divisible by
    2, i.e., 1, 3, 5, 7, and 9. In the test case digits(235), the odd digits are 2, 3, and 5.

def digits(n):
    """
    Given a positive integer n, return the product of the odd digits.
    Return 0 if all digits are even.

    Args:
        n (int): A positive integer.

    Returns:
        int: The product of the odd digits, or 0 if all digits are even.
    """
    # Convert the integer to a string to easily iterate over each digit
    str_n = str(n)

    # Initialize the product to 1
    product = 1

    # Initialize a flag to track if any odd digits were found
    found_odd = False

    # Iterate over each character (digit) in the string
    for char in str_n:
        # Convert the character back to an integer
        digit = int(char)

        # Check if the digit is odd
        if digit % 2 != 0:
            # If the digit is odd, multiply it with the current product
            product *= digit
            # Set the flag to True
            found_odd = True

    # If no odd digits were found, return 0
    if not found_odd:
        return 0
    # Otherwise, return the product of the odd digits
    else:
        return product
```

Listing 6: The output of SEK from Humaneval (problem ID 131)

Despite expanding the search space, Beam Search is still less effective than SEK due to its failure to deepen its understanding of the problem. To illustrate this, we use Problem 131 from Humaneval, generated by Llama 3.1-70B-Instruct. Although Beam Search and the Default implementations differ, neither approach fully comprehends the problem. Specifically, when handling *odd digits*, both methods incorrectly return 0 when the product of the *odd digits* is 1. In contrast, SEK not only identifies but also correctly interprets the concept of *odd digits* in the problem description, allowing it to handle cases where the product of the odd digits equals 1 accurately. This demonstrates that SEK, by focusing on the underlying semantic understanding of key problem concepts, develops a deeper comprehension of the task, ultimately leading to the correct solution.

## J   MORE EXAMPLES

We present an example from APPS-Introductory. This problem requires solving an optimization problem to minimize the number of thieves while ensuring that each thief steals the same number of swords and accounting for all missing swords. This solution requires the use of the greatest common divisor (GCD). It can be seen that both CoT and Default make the same mistake: they use the difference between the maximum and minimum sword counts to calculate the number of swords each person takes. In contrast, only SEK correctly utilizes the GCD, resulting in the generation of accurate code.

```
There were $n$ types of swords in the theater basement which had been used during the plays. Moreover there
    were exactly $x$ swords of each type. $y$ people have broken into the theater basement and each of them
    has taken exactly $z$ swords of some single type. Note that different people might have taken different
    types of swords. Note that the values $x, y$ and $z$ are unknown for you.

The next morning the director of the theater discovers the loss. He counts all swords | exactly $a_i$ swords
    of the $i$-th type are left untouched.

The director has no clue about the initial number of swords of each type in the basement, the number of
    people who have broken into the basement and how many swords each of them have taken.

For example, if $n=3$, $a = [3, 12, 6]$ then one of the possible situations is $x=12$, $y=5$ and $z=3$. Then
    the first three people took swords of the first type and the other two people took swords of the third
    type. Note that you don't know values $x, y$ and $z$ beforehand but know values of $n$ and $a$.

Thus he seeks for your help. Determine the minimum number of people $y$, which could have broken into the
    theater basement, and the number of swords $z$ each of them has taken.

-----Input-----

The first line of the input contains one integer $n$ $(2 \le n \le 2 \cdot 10^{5})$ | the number of types of
    swords.

The second line of the input contains the sequence $a_1, a_2, \dots, a_n$ $(0 \le a_i \le 10^{9})$, where
    $a_i$ equals to the number of swords of the $i$-th type, which have remained in the basement after the
    theft. It is guaranteed that there exists at least one such pair of indices $(j, k)$ that $a_j \neq
    a_k$.

-----Output-----

Print two integers $y$ and $z$ | the minimum number of people which could have broken into the basement and
    the number of swords each of them has taken.

-----Examples-----
Input
3
3 12 6

Output
5 3

Input
2
2 9

Output
1 7

Input
7
2 1000000000 4 6 8 4 2

Output
2999999987 2

Input
6
13 52 0 13 26 52

Output
12 13

-----Note-----
In the first example the minimum value of $y$ equals to $5$, i.e. the minimum number of people who could have
    broken into the basement, is $5$. Each of them has taken $3$ swords: three of them have taken $3$
    swords of the first type, and two others have taken $3$ swords of the third type.
In the second example the minimum value of $y$ is $1$, i.e. the minimum number of people who could have
    broken into the basement, equals to $1$. He has taken $7$ swords of the first type.
```

Listing 7: A problem from APPS-Introductory (problem ID 4372)

```
Below is a Python script with a self-contained function that solves the problem and passes corresponding
    tests:
```

```python
import math

# Read the number of types of swords
n = int(input())
# Read the sequence of remaining swords
a = list(map(int, input().split()))

# Find the minimum and maximum values in the sequence
min_swords = min(a)
max_swords = max(a)

# Calculate the number of swords each person took
z = max_swords - min_swords

# Calculate the minimum number of people
y = 0
if z != 0:
    for swords in a:
        y += (swords - min_swords) // z

# Output the result
print(y, z)
```

This script reads the input, calculates the minimum and maximum values of the remaining swords, and then computes the number of people (`y`) and the number of swords each person took (`z`). It ensures that the output is the minimum number of people who could have broken into the basement and the number of swords each of them has taken.

Listing 8: The output of Default from APPS-Introductory (problem ID 4372)

Below is a Python script with a self-contained function that solves the problem and passes corresponding tests:
```python
# Read the number of types of swords
n = int(input())
# Read the sequence of remaining swords
a = list(map(int, input().split()))

# Step 1: Find the minimum and maximum values in the array
min_swords = min(a)
max_swords = max(a)

# Step 2: Calculate the number of swords each person took
z = max_swords - min_swords

# Step 3: Calculate the minimum number of people
y = 0
for swords in a:
    if swords != max_swords:
        y += (max_swords - swords) // z

# Step 4: Print the result
print(y, z)
```

This script reads the input, calculates the minimum number of people (`y`) and the number of swords each person took (`z`), and prints the result. The logic is based on finding the difference between the maximum and minimum values in the array `a`, which represents the number of swords each person took (`z`), and then summing up the number of people required to achieve this difference for each type of sword.

Listing 9: The output of CoT from APPS-Introductory (problem ID 4372)

There were $n$ types of swords in the theater basement which had been used during the plays. Moreover there were exactly $x$ swords of each type. $y$ people have broken into the theater basement and each of them has taken exactly $z$ swords of some single type. Note that different people might have taken different types of swords. Note that the values $x, y$ and $z$ are unknown for you.

The next morning the director of the theater discovers the loss. He counts all swords | exactly $a_i$ swords of the $i$-th type are left untouched.

The director has no clue about the initial number of swords of each type in the basement, the number of people who have broken into the basement and how many swords each of them have taken.

For example, if $n=3$, $a = [3, 12, 6]$ then one of the possible situations is $x=12$, $y=5$ and $z=3$. Then the first three people took swords of the first type and the other two people took swords of the third type. Note that you don't know values $x, y$ and $z$ beforehand but know values of $n$ and $a$.

Thus he seeks for your help. Determine the minimum number of people $y$, which could have broken into the theater basement, and the number of swords $z$ each of them has taken.

-----Input-----

The first line of the input contains one integer $n$ $(2 \le n \le 2 \cdot 10^{5})$ | the number of types of swords.

The second line of the input contains the sequence $a_1, a_2, \dots, a_n$ $(0 \le a_i \le 10^{9})$, where $a_i$ equals to the number of swords of the $i$-th type, which have remained in the basement after the theft. It is guaranteed that there exists at least one such pair of indices $(j, k)$ that $a_j \neq a_k$.

-----Output-----

Print two integers $y$ and $z$ | the minimum number of people which could have broken into the basement and the number of swords each of them has taken.

-----Examples-----
Input
3
3 12 6

Output
5 3

Input
2
2 9

Output
1 7

Input
7
2 1000000000 4 6 8 4 2

Output
2999999987 2

Input
6
13 52 0 13 26 52

Output
12 13

-----Note-----
In the first example the minimum value of $y$ equals to $5$, i.e. the minimum number of people who could have broken into the basement, is $5$. Each of them has taken $3$ swords: three of them have taken $3$ swords of the first type, and two others have taken $3$ swords of the third type.
In the second example the minimum value of $y$ is $1$, i.e. the minimum number of people who could have broken into the basement, equals to $1$. He has taken $7$ swords of the first type.

-----Keywords and Explanations-----
Analyze the following key terms and their relationships within the problem context:
[number of swords taken]: The quantity of swords, denoted as \( z \), that each person took. This value is derived from the GCD of the differences between the initial number of swords and the remaining number of swords for each type. In the first example, \( z = 3 \) is the result of calculating the GCD of the differences in the remaining swords.
[minimum number of people]: The smallest count of individuals, denoted as \( y \), who could have broken into the theater basement and taken swords. This value is determined by the differences in the number of swords left of each type, \( a_i \), and is calculated as the greatest common divisor (GCD) of these differences. In the first example, \( y = 5 \) is derived from the differences between the remaining swords of each type.

Listing 10: The enriched problem of APPS-Introductory (problem ID 4372)

```python
Below is a Python script that solves the problem described:

import math
from functools import reduce

def gcd(a, b):
    while b:
        a, b = b, a % b
    return a

def find_gcd(list):
    x = reduce(gcd, list)
    return x

n = int(input())
a = list(map(int, input().split()))

# Calculate the differences between the maximum and each element in a
differences = [max(a) - ai for ai in a]

# Find the GCD of all differences to determine the number of swords each person took (z)
z = find_gcd(differences)

# Calculate the minimum number of people (y) by summing up the differences and dividing by z
y = sum(differences) // z

print(y, z)
```

```
```

This script reads the number of types of swords `n` and the list of remaining swords `a` from the input. It
    then calculates the differences between the maximum number of swords and each type's remaining swords.
    The greatest common divisor (GCD) of these differences is computed to find the number of swords each
    person took (`z`). Finally, it calculates the minimum number of people (`y`) by summing up the
    differences and dividing by `z`. The results are printed as `y` and `z`.
```

Listing 11: The output of SEK from APPS-Introductory (problem ID 4372)

```
Write a function to find the difference of the first even and first odd number of a given list.
assert diff_even_odd([1,3,5,7,4,1,6,8])==3
```

Listing 12: A problem from MBPP (problem ID 594)

```
Here is the completed function:

```python
def diff_even_odd(lst):
    first_even = None
    first_odd = None

    for num in lst:
        if num % 2 == 0 and first_even is None:
            first_even = num
        elif num % 2 != 0 and first_odd is None:
            first_odd = num

        if first_even is not None and first_odd is not None:
            break

    if first_even is None or first_odd is None:
        return None

    return abs(first_even - first_odd)

# Test case
assert diff_even_odd([1, 3, 5, 7, 4, 1, 6, 8]) == 3
```
```

Listing 13: The output of Default from MBPP (problem ID 594)

```
Certainly! Below is a self-contained Python script that defines a function to find the difference between the
    first even and first odd number of a given list, along with a corresponding test:

```python
def diff_even_odd(numbers):
    first_even = None
    first_odd = None

    for number in numbers:
        if number % 2 == 0 and first_even is None:
            first_even = number
        elif number % 2 != 0 and first_odd is None:
            first_odd = number

        if first_even is not None and first_odd is not None:
            break

    if first_even is None or first_odd is None:
        return None  # Return None if either an even or odd number is not found

    return abs(first_even - first_odd)

# Test the function
assert diff_even_odd([1, 3, 5, 7, 4, 1, 6, 8]) == 3
```

This script defines the `diff_even_odd` function, which iterates through the given list to find the first
    even and first odd numbers. It then calculates the absolute difference between these two numbers and
    returns it. The test case provided in the problem statement is also included to verify the correctness
    of the function.
```

Listing 14: The output of CoT from MBPP (problem ID 594)

```
Write a function to find the difference of the first even and first odd number of a given list.
assert diff_even_odd([1,3,5,7,4,1,6,8])==3

Analyze the following key terms and their relationships within the problem context:
    [even number]: An integer that is divisible by 2 without leaving a remainder. In the provided list
        `[1,3,5,7,4,1,6,8]`, the first even number is `4`.
[odd number]: An integer that is not divisible by 2, leaving a remainder of 1. In the provided list
    `[1,3,5,7,4,1,6,8]`, the first odd number is `1`.
```

```
[difference]: The mathematical operation of subtracting one number from another. In the context of the
    function, it refers to the result of subtracting the first odd number from the first even number in the
    given list. For the test case `diff_even_odd([1,3,5,7,4,1,6,8])==3`, the difference is calculated as `4
    - 1 = 3`.

assert diff_even_odd([1,3,5,7,4,1,6,8])==3
```

Listing 15: The enriched problem of MBPP (problem ID 594)

```
Here is the self-contained Python script that solves the problem described:

```python
def diff_even_odd(numbers):
    """
    Find the difference of the first even and first odd number of a given list.

    Args:
    numbers (list): A list of integers.

    Returns:
    int: The difference between the first even and first odd number.
    """
    first_even = None
    first_odd = None

    for num in numbers:
        if first_even is None and num % 2 == 0:
            first_even = num
        if first_odd is None and num % 2 != 0:
            first_odd = num

        if first_even is not None and first_odd is not None:
            break

    if first_even is None or first_odd is None:
        raise ValueError("List must contain at least one even and one odd number.")

    return first_even - first_odd

# Test case
assert diff_even_odd([1, 3, 5, 7, 4, 1, 6, 8]) == 3
```

This script defines a function `diff_even_odd` that takes a list of integers as input and returns the
    difference between the first even number and the first odd number in the list. The function iterates
    through the list to find the first even and first odd numbers and then calculates their difference. The
    test case provided in the problem statement is also included to verify the function's correctness.
```

Listing 16: The output of SEK from MBPP (problem ID 594)

# K DETAILS OF DEMONSTRATIONS USED IN KEYEXTRACT & EXPLAIN

```
Demonstration 1:
Check if in given list of numbers, are any two numbers closer to each other than given threshold.
    >>> has_close_elements([1.0, 2.0, 3.0], 0.5)
    False
    >>> has_close_elements([1.0, 2.8, 3.0, 4.0, 5.0, 2.0], 0.3)
    True

[closer to each other]: Describes two numbers in the list whose absolute difference is less than the given
    threshold. For example, in the list [1.0, 2.8, 3.0, 4.0, 5.0, 2.0] with a threshold of 0.3, the numbers
    2.8 and 3.0 are considered closer to each other because |2.8 - 3.0| = 0.2, which is less than 0.3.
[has_close_elements]: Function name that defines the operation to be implemented. It takes two arguments: a
    list of numbers and a threshold value. The function should return True if any two numbers in the list
    have a difference smaller than the threshold, and False otherwise.

Demonstration 2:
Input to this function is a string containing multiple groups of nested parentheses. Your goal is to separate
    those group into separate strings and return the list of those.
    Separate groups are balanced (each open brace is properly closed) and not nested within each other
    Ignore any spaces in the input string.
    >>> separate_paren_groups('( ) (( )) (( )( ))')
    ['()', '(())', '(()())']

[balanced]: Refers to parentheses groups where each opening parenthesis '(' has a corresponding closing
    parenthesis ')' in the correct order, without any mismatches. Examples of balanced groups include '()',
    '(())', and '(()())'. In a balanced group, the number of opening and closing parentheses is always
    equal.
[nested parentheses]: Describes parentheses groups where complete inner pairs are fully contained within
    outer pairs, without overlapping. The group '(()())' demonstrates this concept, containing two complete
    inner pairs '()' nested within an outer pair. Nested groups can have multiple levels of nesting while
    still being balanced.
[separate_paren_groups]: Function name indicating the functionality to be implemented. This function takes a
    single string argument containing multiple groups of nested parentheses. It should return a list of
    separated, independent parentheses groups.
```

Listing 17: The selected demonstrations and corresponding keywords and explanations in Humaneval(+) benchmark

```
Write a function to find the shared elements from the given two lists.
assert set(similar_elements((3, 4, 5, 6),(5, 7, 4, 10))) == set((4, 5))

[shared elements]: Elements that appear in both input lists or sequences. In the test case, 4 and 5 are the
    shared elements between (3, 4, 5, 6) and (5, 7, 4, 10), as they occur in both sequences.
[similar_elements]: Function name indicating the operation to be implemented. It takes two lists (or tuples)
    as input and should return a collection of elements common to both input sequences.
```

Listing 18: The selected demonstrations and corresponding keywords and explanations in MBPP(+)

```
Demonstration 1:
You have $n$ barrels lined up in a row, numbered from left to right from one. Initially, the $i$-th barrel
    contains $a_i$ liters of water.

You can pour water from one barrel to another. In one act of pouring, you can choose two different barrels
    $x$ and $y$ (the $x$-th barrel shouldn't be empty) and pour any possible amount of water from barrel
    $x$ to barrel $y$ (possibly, all water). You may assume that barrels have infinite capacity, so you can
    pour any amount of water in each of them.

Calculate the maximum possible difference between the maximum and the minimum amount of water in the barrels,
    if you can pour water at most $k$ times.

Some examples:  if you have four barrels, each containing $5$ liters of water, and $k = 1$, you may pour $5$
    liters from the second barrel into the fourth, so the amounts of water in the barrels are $[5, 0, 5,
    10]$, and the difference between the maximum and the minimum is $10$;  if all barrels are empty, you
    can't make any operation, so the difference between the maximum and the minimum amount is still $0$.

-----Input-----

The first line contains one integer $t$ ($1 \le t \le 1000$) | the number of test cases.

The first line of each test case contains two integers $n$ and $k$ ($1 \le k < n \le 2 \cdot 10^5$) | the
    number of barrels and the number of pourings you can make.

The second line contains $n$ integers $a_1, a_2, \dots, a_n$ ($0 \le a_i \le 10^{9}$), where $a_i$ is the
    initial amount of water the $i$-th barrel has.

It's guaranteed that the total sum of $n$ over test cases doesn't exceed $2 \cdot 10^5$.

-----Output-----

For each test case, print the maximum possible difference between the maximum and the minimum amount of water
    in the barrels, if you can pour water at most $k$ times.

-----Example-----
Input
2
4 1
5 5 5 5
3 2
0 0 0

Output
10
0

[barrels]: Containers numbered from 1 to n, where the i-th barrel initially contains a_i liters of water. In
    the first example, there are 4 barrels, each containing 5 liters of water, represented as [5, 5, 5, 5].
[maximum difference]: The largest possible gap between the fullest and emptiest barrels after performing up
    to k pourings. For the first example, this value is 10, achieved by creating a barrel with 10 liters
    and another with 0 liters.

Demonstration 2:
Mikhail walks on a Cartesian plane. He starts at the point $(0, 0)$, and in one move he can go to any of
    eight adjacent points. For example, if Mikhail is currently at the point $(0, 0)$, he can go to any of
    the following points in one move:  $(1, 0)$;  $(1, 1)$;  $(0, 1)$;  $(-1, 1)$;  $(-1, 0)$;  $(-1,
    -1)$;  $(0, -1)$;  $(1, -1)$.

If Mikhail goes from the point $(x1, y1)$ to the point $(x2, y2)$ in one move, and $x1 \ne x2$ and $y1 \ne
    y2$, then such a move is called a diagonal move.

Mikhail has $q$ queries. For the $i$-th query Mikhail's target is to go to the point $(n_i, m_i)$ from the
    point $(0, 0)$ in exactly $k_i$ moves. Among all possible movements he want to choose one with the
    maximum number of diagonal moves. Your task is to find the maximum number of diagonal moves or find
    that it is impossible to go from the point $(0, 0)$ to the point $(n_i, m_i)$ in $k_i$ moves.

Note that Mikhail can visit any point any number of times (even the destination point!).
```

```
-----Input-----

The first line of the input contains one integer $q$ ($1 \le q \le 10^4$) | the number of queries.

Then $q$ lines follow. The $i$-th of these $q$ lines contains three integers $n_i$, $m_i$ and $k_i$ ($1 \le
    n_i, m_i, k_i \le 10^{18}$) | $x$-coordinate of the destination point of the query, $y$-coordinate of
    the destination point of the query and the number of moves in the query, correspondingly.

-----Output-----

Print $q$ integers. The $i$-th integer should be equal to -1 if Mikhail cannot go from the point $(0, 0)$ to
    the point $(n_i, m_i)$ in exactly $k_i$ moves described above. Otherwise the $i$-th integer should be
    equal to the the maximum number of diagonal moves among all possible movements.

-----Example-----
Input
3
2 2 3
4 3 7
10 1 9

Output
1
6
-1

-----Note-----

One of the possible answers to the first test case: $(0, 0) \to (1, 0) \to (1, 1) \to (2, 2)$.

One of the possible answers to the second test case: $(0, 0) \to (0, 1) \to (1, 2) \to (0, 3) \to (1, 4) \to
    (2, 3) \to (3, 2) \to (4, 3)$.

In the third test case Mikhail cannot reach the point $(10, 1)$ in 9 moves.

[revisiting]: The ability to pass through any point, including the destination, multiple times during the
    journey. In the second example (4, 3, 7), the optimal path includes revisiting coordinates: (0, 0) →
    (0, 1) → (1, 2) → (0, 3) → (1, 4) → (2, 3) → (3, 2) → (4, 3). This feature allows for maximizing
    diagonal moves even when the direct path wouldn't utilize all available moves.
```

Listing 19: The selected demonstrations and corresponding keywords and explanations in APPS

