# OpenReview forum: "Self-Explained Keywords Empower Large Language Models for Code Generation"
_ICLR.cc/2025/Conference — Submitted to ICLR 2025_

### Official Review · Reviewer_t5jd · 2024-10-29

**Soundness:** 2
**Presentation:** 3
**Contribution:** 2
**Rating:** 3
**Confidence:** 4

**Summary:**

This paper introduces **SEK** (Self-Explained Keywords), a pipeline to improve large language model (LLM) code generation by translating low-frequency keywords in the problem description into high-frequency natural language descriptions. The authors evaluate the effectiveness of SEK on three code generation benchmarks and claim that SEK provides substantial and consistent performance improvements.

**Strengths:**

- This paper presents a well-designed pipeline to address the issue of overlooking low-frequency terms in program descriptions due to the long-tail distribution present in LLM training data. The experimental results demonstrate that SEK effectively enhances code generation performance.

- The authors have conducted a wide spectrum of experiments, encompassing five leading LLMs, four baseline models, and three established code generation benchmarks. This extensive evaluation adds robustness and credibility to the findings.

**Weaknesses:**

- **Simplistic Benchmarks:** The selected benchmarks seem relatively simple and may not adequately capture the real-world effectiveness of the proposed approach. To enhance the rigor and applicability of this study, incorporating more recent and realistic benchmarks [1,2] would be beneficial. This would strengthen the overall soundness and relevance of the paper.

- **Similarity to One-step CoT or One-shot Learning:** The SEK approach exhibits similarities with one-step chain-of-thought (CoT) and one-shot learning strategies. To better elucidate and highlight the advantages of SEK, I suggest conducting a simple experiment, which could ask a language model to rephrase the problem description using precise language. The rephrased description would then be fed back into the language model to determine if this simple rephrasing enhances performance as well as SEK does.

[1] Zhuo, T. Y., Vu, M. C., Chim, J., Hu, H., Yu, W., Widyasari, R., ... & Von Werra, L. (2024). Bigcodebench: Benchmarking code generation with diverse function calls and complex instructions. arXiv preprint arXiv:2406.15877.

[2] Shypula, A., Madaan, A., Zeng, Y., Alon, U., Gardner, J., Hashemi, M., ... & Yazdanbakhsh, A. (2023). Learning performance-improving code edits. arXiv preprint arXiv:2302.07867.

**Questions:**

See Weaknesses.

---

> ### Author Response · Authors · 2024-11-21
> **Official Comment by Authors (1/2)**
>
> > W1: Simplistic Benchmarks
>
> Thank you for your comment. We respectfully disagree with the assessment that our datasets are simplistic and do not adequately capture the real-world effectiveness. It is worth mentioning that Automated Programming Progress Standard (APPS) [1], is a comprehensive benchmark that includes problems from various coding platforms such as Codeforces and Kattis. It includes three categories, namely Introductory, Interview and Competition, and the Interview-level problems are challenging, matching the difficulty of technical programming interviews. Additionally, HumanEval and MBPP are currently the most widely adopted benchmarks in the code generation field [2,3,4,5].
>
> While the suggested benchmarks are valuable contributions, they serve different purposes or have temporal limitations. The Performance-Improving Edits benchmark [6] focuses on code optimization rather than code generation, which falls outside the scope of our study. As for BigCodeBench [7], while interesting, it is currently under review at ICLR 2025 and has not yet been peer-reviewed or officially published. Furthermore, given its recent release (June 18, 2024), it has limited community validation and adoption (26 citations on arXiv) compared to our selected benchmarks.  We believe our chosen benchmarks provide a well-established and appropriate framework for evaluating our method's effectiveness in code generation tasks.
>
> [1] Hendrycks D, Basart S, Kadavath S, et al. Measuring Coding Challenge Competence With APPS[C]//Thirty-fifth Conference on Neural Information Processing Systems Datasets and Benchmarks Track (Round 2).
>
> [2] Chen B, Zhang F, Nguyen A, et al. CodeT: Code Generation with Generated Tests[C]//The Eleventh International Conference on Learning Representations.
>
> [3] Luo Z, Xu C, Zhao P, et al. WizardCoder: Empowering Code Large Language Models with Evol-Instruct[C]//The Twelfth International Conference on Learning Representations.
>
> [4] Zhang T, Yu T, Hashimoto T, et al. Coder reviewer reranking for code generation[C]//International Conference on Machine Learning. PMLR, 2023: 41832-41846.
>
> [5] Nguyen A, Karampatziakis N, Chen W. Meet in the middle: A new pre-training paradigm[J]. Advances in Neural Information Processing Systems, 2024, 36.
>
> [6]Shypula, A., Madaan, A., Zeng, Y., Alon, U., Gardner, J., Hashemi, M., ... & Yazdanbakhsh, A. (2023). Learning performance-improving code edits. arXiv preprint arXiv:2302.07867.
>
> [7] Zhuo, T. Y., Vu, M. C., Chim, J., Hu, H., Yu, W., Widyasari, R., ... & Von Werra, L. (2024). Bigcodebench: Benchmarking code generation with diverse function calls and complex instructions. arXiv preprint arXiv:2406.15877.
>
> ---
> ###### To be continued.

---

> > ### Author Response · Authors · 2024-11-21
> > **Official Comment by Authors (2/2)**
> >
> > > W2: Similarity to One-step CoT or One-shot Learning
> >
> > Thank you for this insightful suggestion. Following your recommendation, we implemented and evaluated the One-Step Chain-of-Thought (CoT) approach where we first prompted the LLM to "Rephrase the problem description using precise language", then used this refined description to guide the final code generation.  To ensure a fair comparison, we maintained the same few-shot settings as used in SEK. We did not implement One-Step CoT with DeepSeekCoder-V2-Instruct as DeepSeek's API for this model has changed and the API we used for evaluation is no longer accessible. Given our limited GPU resources, we were unable to deploy this model (236B) locally. As shown in Table 1 and Table 6 in the revised paper, One-Step CoT generally performs worse than SEK across most cases. Upon manual inspection of the generated problem descriptions, we identified that LLMs, without human intervention, often struggle to consistently produce precise whole-problem reformulations. Any errors in this intermediate generation step can compromise the overall description accuracy. In contrast, SEK's approach of analyzing specific keywords within the problem description helps mitigate the potential errors that might arise from whole-problem reformulation. These findings further validate the effectiveness of our SEK methodology in enhancing code generation performance.  For clear reading, we present the results of One-Step CoT baseline.
> >
> > | Model                       | Method       | HumanEval | HumanEval+ | MBPP     | MBPP+    | APPS Introductory | APPS Interview | APPS Competition | Average  |
> > | --------------------------- | ------------ | --------- | ---------- | -------- | -------- | ----------------- | -------------- | ---------------- | -------- |
> > | Llama-3.1-70B-Instruct      | Default      | 78.0      | 73.8       | 87.6     | 70.9     | 50.0              | 15.0           | 5.0              | 54.3     |
> > |                             | One-Step CoT | 79.3      | 73.2       | 71.7     | 57.4     | 50.0              | 17.2           | 3.3              | 50.3     |
> > |                             | SEK          | **84.8**  | **79.3**   | **88.4** | **71.2** | **61.7**          | **20.0**       | **8.3**          | **59.1** |
> > | Mixtral-8×22B-Instruct-v0.1 | Default      | 76.2      | 72.0       | 73.8     | 64.3     | 28.3              | 7.7            | 1.6              | 46.3     |
> > |                             | One-Step CoT | 72.0      | 66.5       | **79.6** | **66.9** | 31.6              | 6.1            | 1.6              | 46.3     |
> > |                             | SEK          | **81.1**  | **75.6**   | 79.1     | **66.9** | **33.3**          | **10.0**       | **6.6**          | **50.4** |
> > | GPT-3.5-turbo (API)         | Default      | 72.6      | 67.7       | **84.1** | 71.2     | 46.6              | 18.3           | 0.0              | 51.5     |
> > |                             | One-Step CoT | 70.1      | 65.9       | 78.6     | 66.1     | **53.3**          | 16.1           | 1.6              | 50.2     |
> > |                             | SEK          | **75.6**  | **69.5**   | **84.1** | **72.5** | **53.3**          | **20.6**       | **5.0**          | **54.4** |
> > | GPT-4o-mini (API)           | Default      | **87.8**  | **84.1**   | 85.7     | 72.8     | 53.3              | 31.6           | 11.6             | 61.0     |
> > |                             | One-Step CoT | 86.0      | 79.3       | 85.4     | 70.9     | 45.0              | 29.4           | 10.0             | 58.0     |
> > |                             | SEK          | 87.2      | **84.1**   | **87.8** | **74.1** | **58.3**          | **35.0**       | **13.3**         | **62.8** |
> >
> >
> > | **Model**              | **Method**   | **Introductory(A)** | **Introductory(B)** | **Introductory(C)** | **Average** |
> > | ---------------------- | ------------ | ------------------- | ------------------- | ------------------- | ----------- |
> > | Llama-3.1-70B-Instruct | Default      | 51.6                | 45.0                | 46.6                | 47.7        |
> > |                        | One-Step CoT | 48.3                | 48.3                | 48.3                | 48.3        |
> > |                        | SEK          | **58.3**            | **56.6**            | **50.0**            | **55.0**    |
> > | GPT-3.5-turbo (API)    | Default      | 45.0                | 51.6                | 43.3                | 46.6        |
> > |                        | One-Step CoT | **53.3**            | 48.3                | 41.6                | 47.7        |
> > |                        | SEK          | 48.3                | **53.3**            | **50.0**            | **50.5**    |

---

> > > ### Author Response · Authors · 2024-11-29
> > > **Kindly Reminder**
> > >
> > > Dear Reviewer t5jd,
> > >
> > > Thank you once again for your time! We understand that you may have busy schedules, and we kindly remind you that the discussion deadline is approaching in several days. We are keen to know if our rebuttal has cleared all misunderstandings and what points still require further explanation. These are crucial for improving the quality of our paper. Thank you for your constructive assistance with our work!
> > >
> > > We eagerly await your further responses.
> > >
> > > Sincerely,
> > >
> > > Authors

---

> > > > ### Comment · Reviewer_t5jd · 2024-12-02
> > > >
> > > > I thank the authors for sharing different perspectives.
> > > >
> > > > While I appreciate your approach, I believe incorporating more challenging benchmarks is crucial. The primary contribution of your work lies in offering more precise explanations for low-frequency keywords. However, the benchmarks you’ve chosen are already well-understood and straightforward, requiring little to no further explanation. This undermines the ability of your experiments to effectively substantiate this key contribution.
> > > >
> > > > Regarding the similarity to one-shot CoT, thank you for the additional explanations and experiments. Interestingly, in most cases, CoT appears to negatively impact model performance in your experiments, which is quite unusual. A potential explanation could be that, in the CoT experiments, you directly used the "refined" descriptions to generate code, whereas in the original experiment, you concatenated the SEK explanations with the descriptions. The "refined" descriptions may have disrupted the LLM’s learned patterns, leading to the observed performance degradation. These findings also raise a broader concern about potential data contamination within these LLMs. In the further version, It would be valuable to delve into why SEK enhances code generation performance, whereas CoT does not. For now, I will maintain my score.

---

> ### Author Response · Authors · 2024-12-03
> **Official Comment by Authors (1/2)**
>
> Thank you for your valuable feedback.
>
> Regarding your concern about the benchmarks, we respectfully disagree with the assessment that our chosen benchmarks are "well-understood and straightforward, requiring little to no further explanation". First, if this is the case, then SEK, which focuses on providing explicit and additional explanations, should perform no better than the baselines. However, our experimental results show SEK consistently achieves superior performance across different LLMs, supporting that the used benchmarks still require further explanation. Second, the pass rates of the Default baseline on the APPS benchmark are comparable to or worse than those on the BigCodeBench benchmark you recommended [1]. These results imply that **the difficulty of APPS is at least comparable to that of BigCodeBench**: the pass rates on APPS-Introductory align well with those on the full BigCodeBench, while APPS-Interview demonstrates similar difficulty levels to BigCodeBench-Hard. Moreover, all LLMs consistently achieve the lowest performance on APPS-Competition, indicating APPS-Competition is more challenging than BigCodeBench. What's more, APPS has been extensively cited (488 times) and validated by numerous studies [3,4,5] in the field of code generation, while BigCodeBench is still under review at ICLR 2025 and its impact (28 cites) and adoption in the research community are yet to be established. Our approach has shown superior performance on the APPS dataset. Therefore, we believe that our existing benchmark framework is sufficient to substantiate SEK's contributions.
>
>
> |                   | Llama-3.1-70B-Instruct | Mixtral-8×22B-Instruct-v0.1 | DeepSeek-Coder-V2-Instruct | GPT-3.5-turbo | GPT-4o-mini |
> | ----------------- | ---------------------- | --------------------------- | -------------------------- | ------------- | ----------- |
> | APPS Introductory | 50.0                   | 28.3                        | 70                         | 46.6          | 53.3        |
> | APPS Interview    | 15.0                   | 7.7                         | 36.1                       | 18.3          | 31.6        |
> | APPS Competition  | 5.0                    | 1.6                         | 10.0                       | 0.0           | 11.6        |
> | BigCodeBench Full | ~49                    | 45.4                        | 54                         | 44.9          | 51.8        |
> | BigCodeBench Hard | 23.6                   | 19.9                        | 29.4                       | 19.9          | 25.3        |
>
> Regarding CoT's negative impact on model performance, we would like to mention that this finding is actually not unusual. Previous work has already demonstrated CoT's inherent unsuitability for generation tasks [2].
>
> Additionally, we would like to point out that we conducted the experiments strictly following your initial review comments:
>
> > “The rephrased description would then be fed back into the language model to determine if this simple rephrasing enhances performance as well as SEK does.”
>
> We have also followed your new suggestion (despite being raised less than 24 hours before the discussion deadline) and conducted additional experiments to connect the refined problem descriptions with the original problem descriptions. Specifically, we add the prefix "Revised Problem:" before the refined problem description. In the table below, Original One-Step CoT uses the revised problem descriptions as input, the New One-Step CoT combines both the refined and original problem descriptions as input.
>
> The results and analysis remain consistent with our findings in Section 4.1. Specifically, One-Step CoT and SEK extract different types of knowledge from the problem description. One-Step CoT tends to simply restate the complete problem description, while SEK emphasizes low-frequency keywords that effectively bridge the knowledge gap between the problem description and the code implementation.  Consequently, One-Step CoT's approach fails to address the knowledge gap between problem descriptions and implementations, resulting in weaker performance compared to SEK.
>
> ---
> ##### To be continued.

---

> ### Author Response · Authors · 2024-12-03
> **Official Comment by Authors (2/2)**
>
> |                             |                     | HumanEval | HumanEval+ | APPS Introductory | APPS Interview | APPS Competition | Average  |
> | --------------------------- | ------------------- | --------- | ---------- | ----------------- | -------------- | ---------------- | -------- |
> | Llama-3.1-70B-Instruct      | Default             | 78.0      | 73.8       | 50.0              | 15.0           | 5.0              | 44.3     |
> |                             | Original One-Step CoT | 79.3      | 73.2       | 50.0              | 17.2           | 3.3              | 44.6     |
> |                             | New One-Step CoT    | 82.3      | 75.6       | 53.3              | 17.2           | **8.3**          | 47.3     |
> |                             | SEK                 | **84.8**  | **79.3**   | **61.7**          | **20.0**       | **8.3**          | **50.8** |
> | Mixtral-8×22B-Instruct-v0.1 | Default             | 76.2      | 72.0       | 28.3              | 7.7            | 1.6              | 37.1     |
> |                             | Original One-Step CoT | 72.0      | 66.5       | 31.6              | 6.1            | 1.6              | 35.5     |
> |                             | New One-Step CoT    | 70.1      | 65.9       | 31.6              | **10.0**       | 1.6              | 35.8     |
> |                             | SEK                 | **81.1**  | **75.6**   | **33.3**          | **10.0**       | **6.6**          | **41.3** |
> | GPT-3.5-turbo (API)         | Default             | 72.6      | 67.7       | 46.6              | 18.3           | 0.0              | 41.0     |
> |                             | Original One-Step CoT | 70.1      | 65.9       | **53.3**              | 16.1           | 1.6              | 41.4     |
> |                             | New One-Step CoT    | **75.6**  | **69.5**   | 50.0              | 20.0           | 1.6              | 43.3     |
> |                             | SEK                 | **75.6**  | **69.5**   | **53.3**          | **20.6**       | **5.0**          | **44.8** |
> | GPT-4o-mini (API)           | Default             | **87.8**  | 84.1       | 53.3              | 31.6           | 11.6             | 53.6     |
> |                             | Original One-Step CoT | 86.0      | 79.3       | 45.0              | 29.4           | 10.0             | 49.9     |
> |                             | New One-Step CoT    | **87.8**  | 82.3       | 50.0              | **35.0**       | 10.0             | 53.0     |
> |                             | SEK                 | 87.2      | **84.1**   | **58.3**          | **35.0**       | **13.3**         | **55.6** |
>
>
>
> We hope our response addresses your concerns and demonstrates the validity of our benchmarks and the differences between One-Step CoT and SEK. We appreciate your suggestions and the opportunity to further clarify and improve our work.
>
> [1] https://bigcode-bench.github.io/
>
> [2] Sprague Z, Yin F, Rodriguez J D, et al. To cot or not to cot? chain-of-thought helps mainly on math and symbolic reasoning[J]. arXiv preprint arXiv:2409.12183, 2024.
>
> [3]Zhang S, Chen Z, Shen Y, et al. Planning with Large Language Models for Code Generation[C]//The Eleventh International Conference on Learning Representations.
>
> [4]Olausson T X, Inala J P, Wang C, et al. Is Self-Repair a Silver Bullet for Code Generation?[C]//The Twelfth International Conference on Learning Representations. 2023.
>
> [5]Xu S, Fu W, Gao J, et al. Is DPO Superior to PPO for LLM Alignment? A Comprehensive Study[C]//Forty-first International Conference on Machine Learning.

---

### Official Review · Reviewer_mjQR · 2024-10-29

**Soundness:** 2
**Presentation:** 2
**Contribution:** 2
**Rating:** 3
**Confidence:** 3

**Summary:**

This paper presents SEK (Self-Explained Keywords), a straightforward approach designed to enhance the code generation capabilities of large language models (LLMs). SEK utilizes the LLM to identify and elucidate keywords from problem descriptions and ranks them based on frequency. The authors conduct extensive experiments to show that SEK aids LLMs in recognizing and clarifying essential concepts within problems, thereby improving the accuracy of generated code solutions.


I concur with the paper's motivation, recognizing that due to the long tail of training data, LLMs often misinterpret or miss problem-specific, low-frequency keywords during code generation, which compromises the accuracy of the generated code. The method outlined in the paper involves three steps: keyword extraction and explanation via prompt-based LLMs, rule-based keyword ranking, and enriched prompt input for the final code generation step.


I maintain reservations about the first step, which depends on the LLM's capability to extract and understand keywords. This reliance on the LLM’s inherent abilities seems contradictory to the paper’s motivation. As noted in the paper, LLMs exhibit biases toward low-frequency text comprehension. Therefore, I remain my concerns for this step. The method needs a more generalized or innovative strategies to mitigate this issue, making it challenging to achieve broad applicability solely with constructed prompts. Have the authors investigated the performance of the LLMs specifically in extracting low-frequency keywords? Is there any observed bias? Given the known instability of LLM results, have the authors performed any experimental analyses or discussions on this issue? For instance, running the LLM multiple times, analyzing variations, and conducting separate experiments on low-frequency words to assess the LLM's effectiveness. I suggest that the authors consider using pre-defined keyword extraction dictionaries or tools alongside LLMs for more robust keyword extraction.


In the second and third steps (keyword ranking and prompt enrichment), the ranking method based on heuristic rules is not very flexible or portable and may become unreliable with updates. The concepts of these steps seem akin to Retrieval Augmented Generation (RAG). I recommend that the author consider enhancing these steps by integrating RAG principles. Using heuristic rules and external low-frequency dictionaries as knowledge sources within RAG could allow for a recombination of LLM and RAG to improve the ranking algorithm. Ultimately, this could enrich the prompt with more relevant retrieved context. I think using RAG may be more effective than relying solely on rule-based ranking because it is closer to current technology trends and makes the paper's approach more flexible.


Overall, although it seems that numerous experiments validate the method's effectiveness, the approach remains fundamentally simple, centered primarily around prompt engineering. It lacks substantial theoretical depth and appears to reiterate existing methods rather than presenting innovative solutions. I recommend that the author consider replacing heuristic rules with RAG and integrating existing keyword extraction tools or custom low-frequency keyword dictionaries to create a more adaptable system. Relying excessively on heuristic rules to enhance prompts could render the method cumbersome and challenging to apply to different datasets or application contexts.

**Strengths:**

a practical motivation for this paper and a good writing in the introduction section.
Extensive experiments in this paper.

**Weaknesses:**

Dependence on LLM's Existing Capabilities: The method heavily relies on the LLM's existing keyword extraction and comprehension abilities, which could perpetuate inherent biases, particularly with low-frequency text.
The method proposed, such as prompt engineering and rule-based ranking, are not fundamentally novel and rely heavily on existing techniques, which may limit their impact in advancing the field.

**Questions:**

1. Did you explore the integration of RAG or similar methods into your approach?
2. Have you considered developing more advanced methodologies or theoretical frameworks at a higher level to enhance your proposed solution?

---

> ### Author Response · Authors · 2024-11-21
> **Official Comment by Authors (1/2)**
>
> > W1: Dependence on LLM's Existing Capabilities, Simplicity, Novelty, and Impact
>
> Thank you for raising these concerns regarding our reliance on LLMs' inherent capabilities and the novelty and impact of our approach.
>
> **Dependence on LLM's Existing Capabilities:** We respectfully argue that the inherent bias in LLMs is limited when it comes to extracting relatively low-frequency terms. In our paper, the concept of low-frequency primarily refers to the frequency with which particular terms are translated into corresponding code, rather than their general occurrence in natural language corpora. LLMs are capable of identifying and explaining these terms due to their extensive training on both general and specialized corpora. To provide a concrete example, consider the term "even digits." When comparing its occurrence in code-related corpora (with approximately 3k mentions of python code searched by GitHub) to general corpora like web data (with around 63k mentions searched by Google), it becomes clear that although the term is less commonly found in code-related contexts, LLMs are still capable of comprehending and extracting these terms effectively due to their training on diverse data sources. Additionally, recent research [1,2] has also demonstrated LLMs' robust capabilities in keyword extraction.
>
> **Simplicity:** We would like to thank you for your encouragement on the effectiveness of our approach in the Summary. Regarding the concern that our approach is simple, we respectfully argue that for an effective approach, simplicity is an advantage instead of a disadvantage. Because it means this approach can be easily implemented and applied to real-world scenarios, and thus can be potentially more impactful than complicated approaches. Examples that support this argument include CoT [3] and Zero-shot-CoT[4], which are also fundamentally simple but are very impactful (attract thousands of citations) and have been widely used in practice.
>
> **Novelty:** While we acknowledge that our approach is simple, we respectfully argue that this does not necessarily mean our approach is not novel.  We would like to emphasize that the novelty of this work is twofold: (1) we demonstrate that explicitly explaining relatively low-frequency terms in problem descriptions benefits LLM-based code generation, which is an insight not explored in previous studies; and (2) our approach strates LLMs' inherent capabilities in a novel way to effectively instantiate this insight and significantly enhances LLMs' code generation performance. We believe SEK demonstrate a fundamental insight: the importance of explaining relatively low-frequency terms in problem descriptions, which has proven effective with comprehensive experiments and provides a solid foundation for more advanced methodologies.
>
> **Impact:** This simplicity and reliance on LLMs' existing capabilities actually strengthens the method's practical applicability - our approach achieves substantial improvements while remaining lightweight and easily deployable.
>
> [1] Maragheh R Y, Fang C, Irugu C C, et al. LLM-take: theme-aware keyword extraction using large language models[C]//2023 IEEE International Conference on Big Data (BigData). IEEE, 2023: 4318-4324.
>
> [2] Lee W, Chun M, Jeong H, et al. Toward keyword generation through large language models[C]//Companion Proceedings of the 28th International Conference on Intelligent User Interfaces. 2023: 37-40.
>
> [3] Wei J, Wang X, Schuurmans D, et al. Chain-of-thought prompting elicits reasoning in large language models[J]. Advances in neural information processing systems, 2022, 35: 24824-24837.
>
> [4] Kojima T, Gu S S, Reid M, et al. Large language models are zero-shot reasoners[J]. Advances in neural information processing systems, 2022, 35: 22199-22213.
>
> > Q1: Integration of RAG
>
> Thank you for suggesting the integration of RAG principles. While RAG is a powerful technology, we deliberately chose not to incorporate it due to its reliance on high-quality knowledge bases and the effectiveness of our proposed approach. First,  integrating RAG or similar methods will make the effectiveness of our approach correlate to the quality of the external knowledge base. Constructing a high-quality knowledge base is non-trivial and may require manual efforts. Moreover, a good knowledge for coding tasks in one domain may not benefit coding tasks in another domain.
>
> In contrast, our approach is self-contained. The heuristics used in the ranking stage do not make any assumptions about the coding task and the model used. Therefore, we respectfully argue our approach is more flexible and portable than RAG-based approaches. In addition, our experimental results have demonstrated that SEK can effectively enhance code generation performance across different models and different benchmarks.  It would be interesting to further investigate whether RAG can benefit the ranking of keywords in future work.
>
> ---
> ###### To be continued.

---

> > ### Comment · Reviewer_mjQR · 2024-11-27
> >
> > Thank you for the authors' response. I partially acknowledge the contributions of this paper. However, I still hold my reservations about the inability of LLMs to handle low-frequency words, so I will keep my rating unchanged.

---

> > > ### Author Response · Authors · 2024-11-27
> > >
> > > Thanks for your response.  Regarding your reservations about the ability of LLMs to hand low-frequency words, we would like to mention that our hypothesis is that **although LLMs may struggle to directly convert low-frequency terms into code, they can identify and explain them.** There are three key aspects that need to be clarified:
> > >
> > > **1. LLMs' coding ability != their ability to identify and explain:** The former is based on code corpora, while the latter relies on larger-scale natural language corpora. Some keywords may be low-frequency in the code dataset but not in the general corpus. For example, "even digits" appears only 3k times in python files (searched by GitHub), but appears 63k times in general web content (indicated by Google).
> > >
> > > **2. LLMs can identify keywords that are low-frequency in code corpus:** Prior work has demonstrated their term extraction capabilities [1,2]. We also include a **frequency distribution analysis** in Appendix E.5, which compares the extracted keywords by LLMs with other terms in the problem descriptions. The results indicate that LLM have the ability to extract relatively low-frequency terms.
> > >
> > > **3. The explanations generated by LLMs can boost code generation:** We have conducted an ablation study by removing the generated explanations while retaining the extracted keywords in Appendix E.6. Following the same experimental setup on HumanEval using Llama-3.1-70B-Instruct and GPT-3.5-turbo, our results show that removing generated explanations leads to performance drops, demonstrating the importance of these explanations and confirming our motivation.
> > >
> > > | Model                  | Method               | Humaneval | Humaneval+ |
> > > | ---------------------- | -------------------- | --------- | ---------- |
> > > | Llama-3.1-70B-Instruct | Default              | 78.0      | 73.8       |
> > > |                        | SEK w/o explanations | 78.7      | 74.4       |
> > > |                        | SEK                  | **84.8**  | **79.3**   |
> > > | GPT-3.5-turbo          | Default              | 72.6      | 67.7       |
> > > |                        | SEK w/o explanations | 72.6      | 68.9       |
> > > |                        | SEK                  | **75.6**  | **69.5**   |
> > >
> > >
> > > #### We hope the clarification and experimental results mentioned above can address your concern about "the inability of LLMs to handle low-frequency words". Or we would really appreciate it if you could provide more information about where we failed to clarify.
> > >
> > > **Reference**
> > >
> > >
> > >
> > > [1] Maragheh R Y, Fang C, Irugu C C, et al. LLM-take: theme-aware keyword extraction using large language models[C]//2023 IEEE International Conference on Big Data (BigData). IEEE, 2023: 4318-4324.
> > >
> > > [2] Lee W, Chun M, Jeong H, et al. Toward keyword generation through large language models[C]//Companion Proceedings of the 28th International Conference on Intelligent User Interfaces. 2023: 37-40.

---

> ### Author Response · Authors · 2024-11-21
> **Official Comment by Authors (2/2)**
>
> > Q2: Developing more advanced methodologies
>
> Yes, we are actively exploring a recursive keyword generation and explanation system to enhance SEK. This framework would iteratively identify and explain keywords until all terms in the problem description are clear enough to be mapped into their implementation, potentially improving code generation performance.
>
> However, we believe our current approach, i.e., SEK, already makes significant contributions by demonstrating a fundamental insight: the importance of explaining relatively low-frequency terms in problem descriptions. This insight has proven effective with comprehensive experiments and provides a solid foundation for more advanced methodologies.
>
> > S1: Considering pre-defined keyword extraction dictionaries or tools alongside LLMs
>
> Thank you for this constructive suggestion. While pre-defined keyword dictionaries can indeed be valuable in certain scenarios, we identified several potential limitations with such an approach. First, given the diversity of models, it is challenging to pre-define an exhaustive dictionary that can cover all possible keywords, and such dictionaries would require frequent updates to remain relevant, potentially limiting their robustness. Second, keywords can vary significantly across domains and contexts.  For example, keywords like "gradient descent" in machine learning or "double-entry bookkeeping" in accounting include specialized phrases that vary across fields. Consequently, this would require maintaining different external knowledge bases tailored to each specific task or domain, which is not only resource-intensive but also inflexible. As a result, we respectfully argue that pre-defined keyword dictionaries might lack the flexibility and portability required for broad applicability.

---

### Official Review · Reviewer_fwP1 · 2024-11-04

**Soundness:** 3
**Presentation:** 3
**Contribution:** 3
**Rating:** 8
**Confidence:** 4

**Summary:**

This paper introduces a two-stage prompting technique called “Self-Explained Keywords”, that improves the quality of code generated from a variety of LLMs. The technique primarily works by first inducing the model to produce descriptions of select keywords, then ranking these descriptions and finally appending them to the original context before proceeding with code generation. The paper suggests that the ideal keywords to select are low-frequency terms in the model training corpus that the model may have more difficulty understanding. The authors evaluate their method on 5 different LLMs across 3 major code generation benchmarks (and an additional variant of the HumanEval and MBPP benchmarks) and find that performance increases across various models.

**Strengths:**

- The paper presents a structured and domain-motivated approach to think about prompt refinement that I think also could be useful in other non-code domains as well. Particularly ones where there is some shared structure between instances or in the general process of solving the task that we can identify apriori.
- It seems possible that one could use this general approach to combine models that are specialized towards explanations with models specialized towards solving tasks in the target domain.
- I think the results point to the fact that many problems may be 'underspecified' and that models are capable of (self) improvement of the specification before solving the problem. The general approach proposed is elegant, simple and targeted.

Overall I think this is an interesting paper, there were a few things that made my score a bit lower that might be addressable by the authors during the discussion.

**Post-discussion update**

I've increased my score based on the updates to the papers from the authors.

**Weaknesses:**

- **No evaluation of zero-shot CoT**: From what I can tell there is no evaluation of zero-shot CoT [Kojima et al, 2022], (aka "lets think step by step"). The proposed method seems similar in spirit to zero-shot CoT albeit more structured and tailored to code generation. The paper would be stronger if it benchmarked against that method, as it would provide a condition that is not dependent on the effect of demonstrations in the original CoT formulation and would allow readers to understand how much problem refinement models are able to do without the use of a specialized prompt.

- **Selecting number of beams=2 because SEK calls the model twice doesn't seem all that well motivated.** Beam search is significantly less costly than full generation so if the goal is to match compute it doesn't seem necessary to limit to just two beams. Since the beam search results are somewhat competititive with other methods, it would be helpful to readers to understand how this saturates as the number of beams increases. The  Wiseman & Rush 2016 citation for beam search experiemnts with beams=5 and beams=10. Could the authors shed some more light on their seelction and why?

- **Results bounds**
	- Table 1 does not show any result bounds like standard deviation or confidence intervals. The authors do present the ranges for the different sampled APPS sets in Table 6 in the appendix, but this should be brought into the main table to allow readers to see the variability of scores for that benchmark.
	- Alternatively the paper would be stronger if it also presented with pass@k (where k > 1) to capture some of the variablity that may be present in each of the methods.
	- In particular the results in Table 4 would benefit from repeated sampling and error bounds to better understand the importance of the ranking step as the scores are somewhat close.

- **Low frequency assumption**
	- One area of the paper that I did not find particularly convincing was the assertion that the keywords worth explaining are low-frequency tokens *in the training corpus*. I couldn't really find any evidence for this presented in the paper, if I missed this then I'd certainly appreciate clarification. If not then I think the paper would be better served if this were framed as a motivating assumption. While I think the intuition that rarity may play a role is not unreasonable, it seems somewhat overstated as causative. To give an example, L043 "The term even digits rarely appears in code datasets, causing LLMs to misinterpret it as even numbers." — how do we know that **"even digits"** occurs rarely in training datasets (presumably compared to "even numbers")?
	- The prompt used to select "keywords" wouldn't necessarily bias towards selecting low frquency terms in the training corpus. It mainly tries to select and expand on key terms for the problem at hand and for generating correct functions

**Questions:**

- Could the authors provide more detail (and an example) on the construction of the full CoT prompts presented to the models (including demonstrations). One surprising result in Table 5 was the large drop in human-eval performance for GPT-3.5-turbo for CoT relative to the default setup. It would be helpful to readers to understand how the demonstrations used might impact quality of generation.
- Table 4 shows results for experiments for 4 different combination orders in order to determine the best one. But shouldn't there be 6? Abs_Func_Gen and Func_Gen_Abs seem to be missing. Why weren't these other combinations evaluated?
	- As an aside it would help to put "SEK (Abs_Gen_Func)" in the table to help the reader compare.
- How many keywords are extracted and how many explanations are generated? The motivation for ranking the keyword descriptions based on locality bias in LLMs is a factor I wouldn't have expected to be relavant over the short token spans of the examples shown so was surprising to see.
- Fig 5 case study. I may have misunderstood something, but I couldn't understand the description of the case study associated with figure 5. It suggests that the CoT result searches for the string "boredom" rather than "I", but I couldn't identify that bug. I transcribed all 3 solutions and found the default and CoT ones were correct and the SEK one was wrong (not correctly capturing the sentences). I've included the code I used below for the authors to examine and point out if i missed something.

```py
def is_bored_default(S):
    # Split the string into sentences based on delimiters '. ', '?', and '!'
    sentences = S.split('.')
    sentences += S.split('?')
    sentences += S.split('!')
    # Remove empty strings from the list
    sentences = [sentence.strip() for sentence in sentences if sentence.strip()]
    # Count the number of sentences that start with "I"
    boredom_count = 0
    for sentence in sentences:
        words = sentence.split()
        if words and words[0].lower() == "i":
            boredom_count += 1
    return boredom_count


def is_bored_cot(S):
    # Step 1: Initialize
    count = 0
    sentences = S.split('.') # Split by '.' first
    # Step 2: Further split by '?' and '!'
    sentences = [sentence for sublist in [sentence.split('? ') for sentence in sentences] for sentence in sublist]
    sentences = [sentence for sublist in [sentence.split('! ') for sentence in sentences] for sentence in sublist]
    # Step 3: Check each sentence
    for sentence in sentences:
        # Step 4: Strip leading/trailing whitespace and ch eck if it starts with "I"
        if sentence.strip().startswith("I"):
            count += 1
    # Step 5: Return the count
    return count


def is_bored_sek(S):
    # Split the string into sentences based on '.', '?', or '!'
    sentences = []
    start = 0
    for i, char in enumerate(S):
        if char in '.?!':
            print("Appending", S[start:i+1].strip())
            sentences.append(S[start:i+1].strip())
            start = i + 1
    # Count the number of boredoms
    boredom_count = 0
    print("Se", sentences)
    for sentence in sentences:
        # Check if the sentence starts with the word "I"
        if sentence.startswith("I ") or sentence == "I":
            boredom_count += 1
    return boredom_count


print("is_bored_default")
print(is_bored_default("Hello world"))
print(is_bored_default("The sky is blue. The sun is shining. I love this weather"))
print(is_bored_default("The sky is blue. I think The sun is shining. I love this weather"))

print("is_bored_cot")
print(is_bored_cot("Hello world"))
print(is_bored_cot("The sky is blue. The sun is shining. I love this weather"))
print(is_bored_cot("The sky is blue. I think The sun is shining. I love this weather"))

print("is_bored_sek")
# Outputs 0
print(is_bored_sek("Hello world"))
# Outputs 0 instead of 1
print(is_bored_sek("The sky is blue. The sun is shining. I love this weather"))
# Outputs 1 instead of 2
print(is_bored_sek("The sky is blue. I think The sun is shining. I love this weather"))
```

---

> ### Author Response · Authors · 2024-11-21
> **Official Comment by Authors (1/4)**
>
> > W1: No evaluation of zero-shot CoT
>
> Thanks for this valuable suggestion. We have conducted additional experiments to incorporate Zero-Shot CoT as a baseline for comparison.  We follow the method outlined in its original paper [1],  which first prompts the LLM to "think step by step" for getting the intermediate reasoning steps and then concatenates the original problem description with the generated intermediate steps as input to get the code solution.  We did not implement Zero-Shot CoT with DeepSeekCoder-V2-Instruct as DeepSeek's API for this model has changed and the API we used for evaluation is no longer accessible. Given our limited GPU resources, we were unable to deploy this model (236B) locally.
>
>  As shown in Table 1 and Table 6 in the revised paper, Zero-Shot CoT consistently underperforms SEK across most scenarios. This may be attributed to the distinct types of the knowledge extracted by the two methods: while Zero-Shot CoT tends to merely restate the complete problem description, SEK focuses on keywords and their explanations, thereby more effectively addressing knowledge gaps during code generation.  The detailed comparison between Zero-Shot CoT and SEK has been added to Section 4.1 in the revised paper. The exceptions where SEK achieves slightly weaker performance than Zero-Shot CoT are all related to the MBPP dataset or GPT-4o-mini. These results are consistent with our original results and were discussed in Section 4.1. For clear reading, we present the results of Zero-Shot CoT baseline below.
>
> | Model                       | Method        | HumanEval | HumanEval+ | MBPP     | MBPP+    | APPS Introductory | APPS Interview | APPS Competition | Average  |
> | --------------------------- | ------------- | --------- | ---------- | -------- | -------- | ----------------- | -------------- | ---------------- | -------- |
> | Llama-3.1-70B-Instruct      | Default       | 78.0      | 73.8       | 87.6     | 70.9     | 50.0              | 15.0           | 5.0              | 54.3     |
> |                             | Zero-Shot CoT | 76.8      | 72.6       | 77.5     | 62,4     | 41.6              | 16.1           | **8.3**          | 48.8     |
> |                             | SEK           | **84.8**  | **79.3**   | **88.4** | **71.2** | **61.7**          | **20.0**       | **8.3**          | **59.1** |
> | Mixtral-8×22B-Instruct-v0.1 | Default       | 76.2      | 72.0       | 73.8     | 64.3     | 28.3              | 7.7            | 1.6              | 46.3     |
> |                             | Zero-Shot CoT | 75.0      | 68.3       | **79.9** | **67.2** | 28.3              | 8.3            | 1.6              | 46.9     |
> |                             | SEK           | **81.1**  | **75.6**   | 79.1     | 66.9     | **33.3**          | **10.0**       | **6.6**          | **50.4** |
> | GPT-3.5-turbo (API)         | Default       | 72.6      | 67.7       | **84.1** | 71.2     | 46.6              | 18.3           | 0.0              | 51.5     |
> |                             | Zero-Shot CoT | 72.6      | 67.1       | 83.3     | 71.2     | 48.3              | **20.6**       | 3.3              | 52.3     |
> |                             | SEK           | **75.6**  | **69.5**   | **84.1** | **72.5** | **53.3**          | **20.6**       | **5.0**          | **54.4** |
> | GPT-4o-mini (API)           | Default       | **87.8**  | 84.1       | 85.7     | 72.8     | 53.3              | 31.6           | 11.6             | 61.0     |
> |                             | Zero-Shot CoT | 86.6      | **84.8**   | **89.7** | **76.2** | 33.3              | 27.2           | 8.3              | 58.0     |
> |                             | SEK           | 87.2      | 84.1       | 87.8     | 74.1     | **58.3**          | **35.0**       | **13.3**         | **62.8** |
>
>
> | Model                  | Method        | Introductory(A) | Introductory(B) | Introductory(C) | Average  |
> | ---------------------- | ------------- | --------------- | --------------- | --------------- | -------- |
> | Llama-3.1-70B-Instruct | Default       | 51.6            | 45.0            | 46.6            | 47.7     |
> |                        | Zero-Shot CoT | 41.6            | 40.0            | 30.0            | 37.2     |
> |                        | SEK           | **58.3**        | **56.6**        | **50.0**        | **55.0** |
> | GPT-3.5-turbo (API)    | Default       | 45.0            | 51.6            | 43.3            | 46.6     |
> |                        | Zero-Shot CoT | **48.3**        | 51.6            | **50.0**        | 50.0     |
> |                        | SEK           | **48.3**        | **53.3**        | **50.0**        | **50.5** |
>
> [1] Kojima T, Gu S S, Reid M, et al. Large language models are zero-shot reasoners[J]. Advances in neural information processing systems, 2022, 35: 22199-22213.
>
> ---
> ###### To be continued.

---

> > ### Comment · Reviewer_fwP1 · 2024-11-23
> >
> > Thanks for your response and updated analysis!

---

> ### Author Response · Authors · 2024-11-21
> **Official Comment by Authors (2/4)**
>
> > W2: On the choice of beam size=2
>
> We appreciate the thoughtful observation regarding our beam search configuration. Our initial selection of beam size=2 was motivated by our aim to compare performance under equivalent search space explorations, as SEK modifies the LLM's search space once through additional token insertion.
>
> To address your concerns about performance saturation and computational costs, we conducted additional experiments with varying beam sizes (2, 3, 5, and 10) using LLaMA-3.1. We were unable to include Mixtral in these experiments due to memory constraints (Out-Of-Memory issues) at beam sizes $\geq$ 5. Our extended results, presented in Table 7 and Table 8 in the revised paper, show that SEK consistently outperforms beam search across most scenarios, even with larger beam sizes. Interestingly, we observed that beam sizes of 5 and 10 occasionally surpassed SEK's performance on MBPP(+) and APPS-Interview, which may be attributed to more computation cost of beam search (see below for details).  For clear reading, we present the results of beam search with different beam sizes.
>
> | Method            | HumanEval | HumanEval+ | MBPP     | MBPP+    | APPS Introductory | APPS Interview | APPS Competition | Average  |
> | ----------------- | --------- | ---------- | -------- | -------- | ----------------- | -------------- | ---------------- | -------- |
> | Default           | 78.0      | 73.8       | 87.6     | 70.9     | 50.0              | 15.0           | 5.0              | 54.3     |
> | Beam Search（2）  | 79.3      | 74.4       | 87.8     | 70.9     | 55.0              | 16.1           | 5.0              | 55.5     |
> | Beam Search（3）  | 78.0      | 74.4       | 87.8     | 72.2     | 53.3              | 20.0           | 6.6              | 56.0     |
> | Beam Search（5）  | 79.9      | 75.6       | 88.4     | **72.8** | 55.0              | **21.1**       | 6.7              | 57.1     |
> | Beam Search（10） | 79.9      | 75.0       | **88.9** | 72.5     | 56.6              | **21.1**       | **8.3**          | 57.5     |
> | SEK               | **84.8**  | **79.3**   | 88.4     | 71.2     | **61.7**          | 20.0           | **8.3**          | **59.1** |
>
> | **Method**        | **Introductory(A)** | **Introductory(B)** | **Introductory(C)** | **Average** |
> | ----------------- | ------------------- | ------------------- | ------------------- | ----------- |
> | Default           | 51.6                | 45.0                | 46.6                | 47.7        |
> | Beam Search（2）  | 55.0                | 45.0                | 45.0                | 48.3        |
> | Beam Search（3）  | 50.0                | 45.0                | 45.0                | 46.7        |
> | Beam Search（5）  | 53.3                | 43.3                | 43.3                | 46.6        |
> | Beam Search（10） | 53.3                | 45.0                | 48.3                | 48.9        |
> | SEK               | **58.3**            | **56.6**            | **50.0**            | **55.0**    |
>
> In practice, especially in LLM application scenarios where computational resources are already heavily constrained, if the GPU memory is large enough to handle $n$ beams, we can also batch $n$ independent user requests and process them simultaneously. However, beam search would require much more computational resources to handle these n requests, as each request needs to maintain multiple beams. So we respectfully argue that the claim "Beam Search is significantly less costly than full generation" may not hold in real-world applications.
>
> To quantify computational resource usage of each approach, we calculated the product of the numbers of generated tokens and maintained paths as the total computational cost. The results are shown in Table 9 and Table 10 in the revised paper. When comparing the scenarios with similar computational costs, SEK consistently outperforms beam search. In the cases where beam search surpasses SEK, beam search typically demands significantly more computational resources. For instance, on MBPP, beam search with sizes 5 and 10 consumed approximately 890 and 1840 computational units respectively, whereas SEK required only 412 units. These results reinforce SEK's efficiency in achieving superior performance. For clear reading, we present the computational costs of different sizes of Beam Search baseline.
>
> ---
> ###### To be continued.

---

> > ### Comment · Reviewer_fwP1 · 2024-11-23
> >
> > Thanks! This is a helpful analysis and addresses my concerns. With regards to computational complexity, I was partly thinking of the ways beam search is able to effectively utilize caching (such as KV cache) to improve its computational efficiency and skip repeating a fair amount of expensive computation. Nevertheless your updates lay out your motivations and working assumptions clearly and interested readers have access to performance numbers of SEK as the number of beams increase.

---

> ### Author Response · Authors · 2024-11-21
> **Official Comment by Authors (3/4)**
>
> | **Method**        | **HumanEval** | **MBPP** | **APPS Introductory** | **APPS Interview** | **APPS Competition** | **Average** |
> | ----------------- | ------------- | -------- | --------------------- | ------------------ | -------------------- | ----------- |
> | Beam Search（2）  | *242.0*       | 378.0    | 202.0                 | *304.0*            | *416.0*              | *308.4*     |
> | Beam Search（3）  | 723.0         | 538.0    | *286.0*               | 435.0              | 611.0                | 518.6       |
> | Beam Search（5）  | 1200.0        | 890.0    | 455.0                 | 685.0              | 1165.0               | 879.0       |
> | Beam Search（10） | 2500.0        | 1840.0   | 960.0                 | 1360.0             | 2410.0               | 1814.0      |
> | SEK               | 450.0         | 412.0    | 273.0                 | 337.0              | 484.0                | 391.2       |
>
> | **Method**        | **Introductory(A)** | **Introductory(B)** | **Introductory(C)** | **Average** |
> | ----------------- | ------------------- | ------------------- | ------------------- | ----------- |
> | Beam Search（2）  | 192.0               | 200.0               | 202.0               | 198.0       |
> | Beam Search（3）  | *281.6*             | *308.0*             | *308.0*             | *299.2*     |
> | Beam Search（5）  | 460.0               | 485.0               | 480.0               | 475.0       |
> | Beam Search（10） | 970.0               | 1050.0              | 950.0               | 990.0       |
> | SEK               | 270.0               | 269.0               | 281.0               | 273.3       |
>
> > W3: Results bounds
>
> Thank you for raising these important points about result bounds and evaluation metrics. We would like to clarify that our experimental setup employs greedy decoding, as mentioned in Section 3.4. Under this configuration, the token generation process is deterministic and repeated runs would yield identical results. These explain the absence of standard deviations or confidence intervals in our presented results.
>
> Regarding the suggestion about pass@k (where k >= 1) evaluation, we note that both our method and the selected baselines are not designed to optimize for multiple attempts. Our focus is on the model's ability to generate correct code solutions in a single pass, so we specifically chose pass@1 as our evaluation metric.
>
> > W4: Low frequency assumption
>
> We appreciate your careful examination of our low-frequency assumption. In response to your feedback, we have revised our Introduction section to present this relationship more as an observation rather than a strict causative relationship. To provide concrete evidence for the comparative frequency of "even digits" versus "even numbers" in programming contexts, we conducted an analysis using GitHub code-specific searches. We observed that "even numbers" appears approximately 215k times, while "even digits" appears only 20k times.
>
> Regarding your point about keyword selection, we respectfully argue that **keywords** and **low-frequency** terms are somewhat similar concepts. According to wikipedia (https://en.wikipedia.org/wiki/Keyword_(linguistics)), a keyword is defined as a word that occurs in a text more often than we would expect by chance alone, based on a comparison between its frequency in a specific text and its expected frequency in a much larger reference corpus. Thus, when we ask the LLM to extract keywords, it is essentially extracting words that are relatively low-frequency in the training set, but appear more frequently in the targeted problem description.
>
> > Q1: Details of CoT prompt
>
> Thank you for the advice. We followed the implementation methodology from the original CoT paper [1], where demonstrations are constructed by combining problem descriptions with step-by-step reasoning processes. We share the complete CoT prompt as follows:
>
> Please provide a self-contained Python script that solves the following problem in a markdown code block:
>
> ```
> """
> Write a function to find the shared elements from the given two lists.
> assert set(similar_elements((3, 4, 5, 6),(5, 7, 4, 10))) == set((4, 5))
> """
>
> Below is a Python script with a self-contained function that solves the problem and passes corresponding tests:
> """python
> def similar_elements(list1, list2):
>     # Step 1, convert to sets
>     set1 = set(list1)
>     set2 = set(list2)
>     # Step 2, find the intersection
>     common_elements = set1.intersection(set2)
>     # Step 3, get results
>     return tuple(common_elements)
> """
>
> Please provide a self-contained Python script that solves the following problem in a markdown code block:
> """
> Write a function to find the kth element in the given array using 1-based indexing.
> assert kth_element([12,3,5,7,19], 2) == 3
> """
>
> Below is a Python script with a self-contained function that solves the problem and passes corresponding tests:
> """python
>
> ```
>
> ---
> ###### To be continued.

---

> > ### Author Response · Authors · 2024-11-21
> > **Official Comment by Authors (4/4)**
> >
> > > Q2: Completeness of combination orders
> >
> > Thank you for this observation regarding the completeness of our combination order experiments. Following your suggestion, we have now conducted additional experiments to include the previously missing combinations (Abs_Func_Gen and Func_Gen_Abs), and the complete results are presented in Table 4. Our comprehensive analysis across all six possible orderings confirms that SEK (Abs_Gen_Func) achieves the best performance among all combinations, validating our choice for the final implementation. We have also updated the table notation to clearly indicate "SEK (Abs_Gen_Func)" for better readability, as you suggested.
> >
> > | **Model**                   | **Combination Order** | **HumanEval** | **HumanEval+** | **Average** |
> > | --------------------------- | --------------------- | ------------- | -------------- | ----------- |
> > | Llama-3.1-70B-Instruct      | Default               | 78.0          | 73.8           | 75.9        |
> > |                             | Func\_Abs\_Gen        | 83.5          | 78.7           | 81.1        |
> > |                             | Func\_Gen\_Abs        | 84.1          | **79.3**       | 81.7        |
> > |                             | Gen\_Func\_Abs        | 84.1          | 78.7           | 81.4        |
> > |                             | Gen\_Abs\_Func        | 84.1          | 78.7           | 81.4        |
> > |                             | Abs\_Func\_Gen        | 84.1          | 78.0           | 81.1        |
> > |                             | SEK(Abs\_Gen\_Func)   | **84.8**      | **79.3**       | **82.1**    |
> > | Mixtral-8×22B-Instruct-v0.1 | Default               | 76.2          | 72.0           | 74.1        |
> > |                             | Func\_Abs\_Gen        | 78.0          | 72.0           | 75.0        |
> > |                             | Func\_Gen\_Abs        | **81.1**      | 75.0           | 78.1        |
> > |                             | Gen\_Func\_Abs        | 78.0          | 72.0           | 75.0        |
> > |                             | Gen\_Abs\_Func        | 76.8          | 71.3           | 74.1        |
> > |                             | Abs\_Func\_Gen        | **81.1**      | **75.6**       | **78.4**    |
> > |                             | SEK(Abs\_Gen\_Func)   | **81.1**      | **75.6**       | **78.4**    |
> >
> > > Q3: The number of keywords and the motivation for ranking
> >
> > Thank you for your question regarding keyword extraction. The number of keywords extracted matches exactly with the number of explanations generated, both of which are at most three. Taking Llama-3.1-70B-Instruct's generation on HumanEval as an example,  we observed 0.68 Abstract keywords, 1.65 General keywords, and 0.57 Functional keywords on average for each problem.
> >
> > While you raise an interesting point about the relevance of locality bias in shorter text spans, our consideration of the keyword order was motivated by the position bias phenomenon in LLMs, as discussed in Section 2.2. Through empirical investigation, we found that the order of different keyword categories indeed impacts the model's problem comprehension.  This may be because the extracted keywords are not completely independent of each other, and ordering them in a certain way, e.g., from integrated concepts to implementation details, can help LLMs better structure the understanding.
> >
> > > Q4: Clarification of case study
> >
> > Thank you for your careful examination of the case study in Figure 5. We would like to clarify the error in the solution generated by CoT. The task requires counting sentences that begin with the word “I" (boredoms). The CoT solution matches any sentence beginning with the character “I", rather than specifically the word “I". This can lead to false positives when sentences begin with words like "In" or “It". Thus the CoT solution is incorrect. To fix this bug, we need to perform the following code change:
> > ```
> > if sentence.strip().startswith("I"): -> if sentence.strip().startswith("I "):
> > ```
> >
> > Upon your feedback, we also re-examined the solution generated by SEK. We acknowledge that it  is plausible but incorrect, because it will not capture the last sentence of the input string if this sentence does not end with ".", "?" or "!". While this solution passes the test cases in HumanEval (based on which we select this case), it does not pass the enhanced test cases in HumanEval+. Although this specific example may not demonstrate perfect generation, the additional examples provided in the appendix section substantiate SEK's effectiveness in improving code generation. Based on your feedback, we have updated the case study example in the main text with a correct one.

---

> > ### Comment · Reviewer_fwP1 · 2024-11-23
> >
> > W4: I should say that I still don't find your argument here convincing, keyword also has the colloquial meaning of a word that is important in its context (or a word of great significance), this is operationalized in corpus linguistics using frequency estimates. _More importantly_ to my reading there isn't _evidence presented_ to support this claim, I think you do have evidence that LLMs are able to extract keywords and provide descriptions for these keywords that are helpful for solving (regardless of what mechanism they may be using to identify these keywords). I don't think the GitHub code-specific search results are representative of the "training data" as this low-frequency assumption asserts. I'm also not sure why you compare against results for "even numbers" when description of even digits in figure 1 doesn't use the word "numbers". Again I think this is a fine motivating assumption/hypothesis, but I don't see sufficient evidence presented to support that this is what the LLM is doing.

---

> ### Author Response · Authors · 2024-11-25
>
> Thank you for your constructive comments. According to your suggestion, we have further restructured the Introduction section to better frame our motivation and hypothesis. We have weakened the claim of rarity and avoided stating it as causative.
>
> Regarding your question about the frequency of "even digits" in training datasets, we conducted an analysis using the Python subset of Stack-V2, which serves as the pretraining data for StarCoder2 [1]. The frequency analysis presented in the table below demonstrates that both "even digit" and "even digits" appear significantly less frequently compared to related terms, empirically supporting our statement about their relative rarity in LLM's training corpora.
>
> |                               | even    | number  | numbers | digit   | digits | even number | even numbers | even digit | even digits |
> | ----------------------------- | ------- | ------- | ------- | ------- | ------ | ----------- | ------------ | ---------- | ----------- |
> | Python Subset of The Stack v2 | 2978878 | 6147317 | 1149304 | 1120256 | 556707 | 42204       | 15951        | 1149       | 832         |
>
>
> To address your concerns regarding the low-frequency nature of the extracted keywords, we have included a new **frequency distribution analysis** comparing the extracted keywords with other terms in the problem descriptions in Appendix E.5. The results demonstrate the extracted keywords are significantly skewed towards higher TF-IDF values, indicating that they are relatively low-frequency terms.
>
>
> [1] Lozhkov A, Li R, Allal L B, et al. Starcoder 2 and the stack v2: The next generation[J]. arXiv preprint arXiv:2402.19173, 2024.

---

> > ### Comment · Reviewer_fwP1 · 2024-11-26
> >
> > Great! Thanks for this additional analysis and figures in the appendix, the low frequency hypothesis is much more convincing with them.
> >
> > Thanks to the authors for their engagement during this discussion phase. I've increased my score to reflect the updates/improvements to the manuscript.

---

> > > ### Author Response · Authors · 2024-11-26
> > >
> > > We are very grateful for your constructive comments and questions, which helped improve the quality of our paper significantly. Thank you very much!

---

### Author Response · Authors · 2024-11-25

# Global comment

We sincerely thank all reviewers for their insightful comments and constructive feedback. We're pleased that they agree our paper offers an `interesting and practical perspective` (Reviewer fwP1, Reviewer mjQR). Our work is recognized for its `extensibility and simplicity` (Reviewer fwP1), `practical motivation and clear writing` (Reviewer mjQR), and `well-designed pipeline and robustness` (Reviewer t5jd).  The reviewers also appreciate the paper’s `extensive experiments` (Reviewer mjQR, Reviewer t5jd). **We believe that our work demonstrates a fundamental insight that explicitly explaining relatively low-frequency keywords in problem descriptions can boost code generation and can inspire future research.**  According to their valuable suggestions, we have made the following revisions to improve our paper:

- We have restructured our Introduction section to better frame our motivation and hypothesis. We also added a new frequency distribution analysis in Appendix E.5, which demonstrates that extracted keywords tend to be relatively low-frequency terms compared to other terms in problem descriptions (*Reviewer fwP1 W4*).
- In Section 2.1 (KeyExtract & Explain), we have clarified the number of keywords and explanations in our explanation of Guideline 4 (*Reviewer fwP1 Q3*).
- We have expanded our baselines by including Zero-Shot CoT and One-Step CoT in Section 3.3, with a detailed discussion in Section 4.1 demonstrating SEK's consistent performance benefits. This enhancement addresses the baseline-related concerns raised by *Reviewer t5jd (W2)* and *Reviewer fwP1 (W1)*.
- We have refined our Beam Search description in Section 3.3 to clarify our focus on equal search space explorations (2 attempts). Additionally, we have included a comprehensive resource-performance analysis in Appendix E.4, showing that SEK consistently outperforms Beam Search with close resource consumption, while Beam Search requires 5-10 times more resources to occasionally surpass SEK (*Reviewer fwP1 W2*).
- We have updated the case study in Section 4.3, as we find it overfits the test suite in HumanEval (based on which we selected the case) (*Reviewer fwP1 Q4*).
- In Appendix E.1, we have added the missing experimental results for Abs_Func_Gen and Func_Gen_Abs combination orders. These additional results further validate our choice of Abs_Gen_Func (*Reviewer fwP1 Q2*).

Next, we will address each reviewer's concerns individually.

---

### Author Response · Authors · 2024-12-02
**Kindly Reminder**

Dear Reviewer mjQR and Reviewer t5jd,

We hope this message finds you well. First and foremost, we would like to sincerely thank you for your valuable feedback and thoughtful comments on our submission. We have carefully considered all the suggestions provided and have updated our manuscript accordingly.

- For Reviewer mjQR: we further clarified the reasons why LLMs have the ability to handle low-frequency words and added related experiments in Appendix E.6.
- For Reviewer t5jd: we explained the value of the selected benchmarks and incorporated the One-step CoT baseline into our revised paper.

As the discussion phase is nearing its conclusion with **only one day remaining**, we would greatly appreciate your feedback on our responses to ensure we have adequately addressed your concerns. Your insights and constructive feedback are invaluable to us, and we are eager to hear from you to further improve our paper. If our responses had addressed your concerns, we would be truly grateful if you could consider re-evaluating your score. We are also happy to respond to any other concerns that might arise.

Thank you again for your time and consideration.




Best regards,

The Authors

---

### Meta-Review · Area_Chair_nLSF · 2024-12-16

**Metareview:**

This paper introduces a novel two-stage prompting method called “Self-Explained Keywords”, aimed at improving code generation quality across a variety of large language models (LLMs). The technique involves: (1) prompting the model to generate descriptions for selected keywords from the problem context, (2) ranking these descriptions, and (3) appending the selected descriptions to the original context before initiating code generation. The approach prioritizes low-frequency keywords from the model’s training corpus, hypothesized to be terms the model struggles to understand. The method is evaluated on five different LLMs across three major code generation benchmarks, including variants of HumanEval and MBPP, showing consistent performance improvements.

However, this work has key limitations. Firstly, the evaluation is restricted to a limited set of benchmarks, raising questions about whether the benchmarks sufficiently validate the method’s underlying assumptions. Secondly, the rationale behind the approach lacks clarity and strong scientific grounding. The hypothesis regarding low-frequency keywords and their explanations is not well substantiated, leaving room for skepticism about why the method succeeds.

**Additional Comments On Reviewer Discussion:**

NA

---

### Decision · Program_Chairs · 2025-01-22

Reject